# Arbitrary waveform AC line filtering applicable to hundreds of volts based on aqueous electrochemical capacitors

Mingmao Wu[1,4], Fengyao Chi[1,4], Hongya Geng[1], Hongyun Ma [1], Miao Zhang[1], Tiantian Gao[1], Chun Li [1] & Liangti Qu[1,2,3]

Filtering capacitor is a necessary component in the modern electronic circuit. Traditional filtering capacitor is often limited by its bulky and rigid configuration and narrow workable scope of applications. Here, an aqueous hybrid electrochemical capacitor is developed for alternating current line filtering with an applicable wide frequency range from 1 to 10,000 Hz. This capacitor possesses an areal specific energy density of 438 $\mu F\, V^2\, cm^{-2}$ at 120 Hz, which to the best of our knowledge is record high among aqueous electrochemical capacitors reported so far. It can convert arbitrary alternating current waveforms and even noises to straight signals. After integration of capacitor units, a workable voltage up to hundreds of volts (e.g., 200 V) could be achieved without sacrificing its filtering capability. The integrated features of wide frequency range and high workable voltage for this capacitor present promise for multi-scenario and applicable filtering capacitors of practical importance.

[1] Department of Chemistry, Tsinghua University, 100084 Beijing, P. R. China. [2] Key Laboratory for Advanced Materials Processing Technology, Ministry of Education of China, Department of Mechanical Engineering, Tsinghua University, Beijing 100084, P. R. China. [3] School of Chemistry and Chemical Engineering, Beijing Institute of Technology, 100081 Beijing, P. R. China. [4] These authors contributed equally: Mingmao Wu, Fengyao Chi. Correspondence and requests for materials should be addressed to L.Q. (email: lqu@tsinghua.edu.cn)

Alternating current (AC) to direct current (DC) convertors are a key component for signal stabilization, which gains interest in fields of renewable power generators that derive energy from sources such as wind, solar, moisture, and friction[1–5]. The ever-increasing demand for AC/DC conversion requires that the filtering capacitor functions within broad frequency and voltage ranges, satisfying needs to smoothen arbitrary waves and easily integrate into miniaturized devices[2,6,7]. However, conventional filtering capacitors such as aluminum electrolytic capacitors (AECs) fail to meet the diversified demands because of low specific capacitances.[2] On the other hand, the bulky size of the conventional filtering capacitors impedes application in the integration and miniaturization of functional and flexible electronic circuits[2,7–10].

Electrochemical capacitors (ECs) with easy engineering, high specific capacitance, excellent rate capability, and frequency adaptability provide a promising candidate for the next-generation filtering devices[11–13]. Up to now, various electrode materials including graphene, carbon nanotubes, conductive polymers, pyrolyzed cellulose, and MXene have been employed to fabricate ECs[12,14–18]. Nevertheless, applicable ECs for practical filtering is still lacking, mostly because of their relatively low working voltage[19]. Generally, aqueous ECs can only work with a low-voltage window of ~0.8–1.0 V. Although organic electrolyte could provide a higher applicable voltage (e.g., 2.5–3 V), the flammability, toxicity, volatility, and complex processing procedures of organic electrolyte are the big concerns for practical application, especially for high-frequency condition[20].

Here, we report a hundreds-of-volts workable AC line filter based on a series-connected configuration of aqueous hybrid electrochemical capacitors (AHECs). One AHEC unit is composed of a positive electrode of acid-pretreated poly(3,4-ethylenedioxythiophene) (PEDOT) film, a negative electrode of electrochemically reduced graphene oxide (ErGO) sheets, and aqueous electrolyte. In this EC, both the stable electrode materials have high conductivities and large porous structures, being in favor of quick electron transport and good electric double-layer capacitive behaviors. And combining with the high ion conductivity of the aqueous electrolyte, the AHEC possesses a low equivalent series resistance of $0.21\,\Omega\,cm^2$, which is smaller than that of AEC. Meanwhile, the AHEC delivers the highest energy density of $438\,\mu F\,V^2\,cm^{-2}$ at 120 Hz among the reported aqueous ECs with a comparable phase angle. Consequently, the large specific energy density and low total impedance of this AHEC enables the efficient smoothing of AC into DC within a wide frequency range from 1 Hz to 10,000 Hz. Furthermore, the rational integration of AHEC units in series allows the device filtering more than hundreds of volts without sacrificing the rate performance and frequency-response properties. This voltage window could meet the requirement of electrical appliance from pint-sized portable devices to large-scale ones. Moreover, the connected AHEC units can smooth arbitrary AC waveforms (e.g., decaying, stair, electrocardiogram, rhombus, ellipse, heart, and violent noise waveforms.) into DC signals, indicative of diversified adaptability and good filtering capability for future practical filtering capacitors.

## Results

**Preparation of electrode materials.** Figure 1a schematically illustrates the preparation procedures of AHEC unit. For preparing PEDOT positive electrode (Supplementary Fig. 1), cellulose membrane with the fibril structure (Supplementary Fig. 2) was firstly put on the graphite foil as the sacrifice hydrophilic porous template for loading PEDOT. The precursor solution of PEDOT (PH1000) was then spin-coated on the surface of the

fibril structure. After dissolution of cellulose membrane by concentrated sulfuric acid, a hierarchically porous PEDOT network was adhered on a graphite foil which acts as the current collector (Fig. 1b–d and Supplementary Fig. 3). Simultaneously, the conductivity of PEDOT was greatly improved (from 0.1 to $1200\,s\,cm^{-1}$) due to acid-enhanced conjugated area of polymer chain (Supplementary Fig. 4)[21]. On the other hand, the ErGO electrodes (Supplementary Fig. 5) were prepared by an electrochemical assisted deposition method on the graphite foil, which realize the synchronous reduction and assembly of GO sheets into a 3D interpenetrating porous structure along the direction of electric field (Fig. 1e, f and Supplementary Fig. 6). Notably, the as-prepared hierarchically porous structure and highly conductive skeleton endow the positive electrode and negative electrode with the quick electrons transport and abundant charge adsorption sites, which are essential for the high-frequency responsive electrochemical process[22]. Finally, the AHEC unit was fabricated by assembling the positive electrode and negative electrode face to face with a separator. The as-prepared AHEC unit shows a compact sandwich structure with an apparent volume of $0.12\,cm^{-3}$ and a mass of 138 mg, which is far smaller and lighter than most commercial AECs ($0.26–9.3\,cm^{-3}$ and 600–12,600 mg) (Fig. 1g).

**Electrochemical performances of single electrode and unit.** Three electrode system was used to investigate the stable potential window and the electrochemical performances of PEDOT electrode and ErGO electrode. For the PEDOT electrode, it exhibits quasi-rectangular cyclic voltammetry (CV) curves and a nearly 100% capacitance retention after 10,000 cycles, indicating its high stability, good rate performances and ideal electric double-layer capacitive behavior within the potential window of 0–0.9 V vs saturated calomel electrode (SCE) (Fig. 2a and Supplementary Fig. 7). Within higher potential window, the capacitance retention slightly decreased to 84% after 10,000 cycles (Supplementary Fig. 8), which may due to the decomposition of electrolyte (Supplementary Fig. 9)[23]. Similarly, potential window of −0.9 to 0 V vs SCE was chosen for ErGO electrode according to the nearly 100% capacitance retention after cycle test (Fig. 2b; Supplementary Figs. 7 and 8). Subsequently, in order to avoid the overcharging of any electrode in one device, the loading of active materials of the positive electrode and negative electrode were adjusted according to the charge balance (Fig. 2c).

$$q_+ = C_{A+} \times \Delta V_+ = q_- = C_{A-} \times \Delta V_- \tag{1}$$

where $q$ is the charges stored in electrode, $C_A$ is the specific areal capacitance, $\Delta V$ is potential amplitude, + and − represent the positive electrode and negative electrode, respectively. Furthermore, based on the individual voltage range of the PEDOT (0–0.9 V vs SCE) and ErGO (−0.9–0 V vs SCE), an optimized voltage window of 0–1.8 V for the AHEC unit was obtained by gradually increasing voltage value in refer to CV curves (Fig. 2d).

A two-electrode configuration AHEC unit was constructed by using PEDOT electrode and ErGO electrode. Its rate capability was firstly evaluated by using CV tests at the scan rates ranging from 10 to $2000\,V\,s^{-1}$ (Supplementary Fig. 10). The CV profiles show a good rectangular shape at the scan rate of $1000\,V\,s^{-1}$ (Fig. 2e), and even at a high scan rate of $2000\,V\,s^{-1}$ (Supplementary Fig. 10f). The remained ideal rectangular shape of CV curve demonstrated the fast electron transport within the electrodes. More importantly, the discharge current density shows a perfect linear relationship with the scan rate in the range of ~10–$1000\,V\,s^{-1}$ (Fig. 2f), better than most reported line-filtering ECs. Apart from the CV tests, the galvanostatic charge–discharge

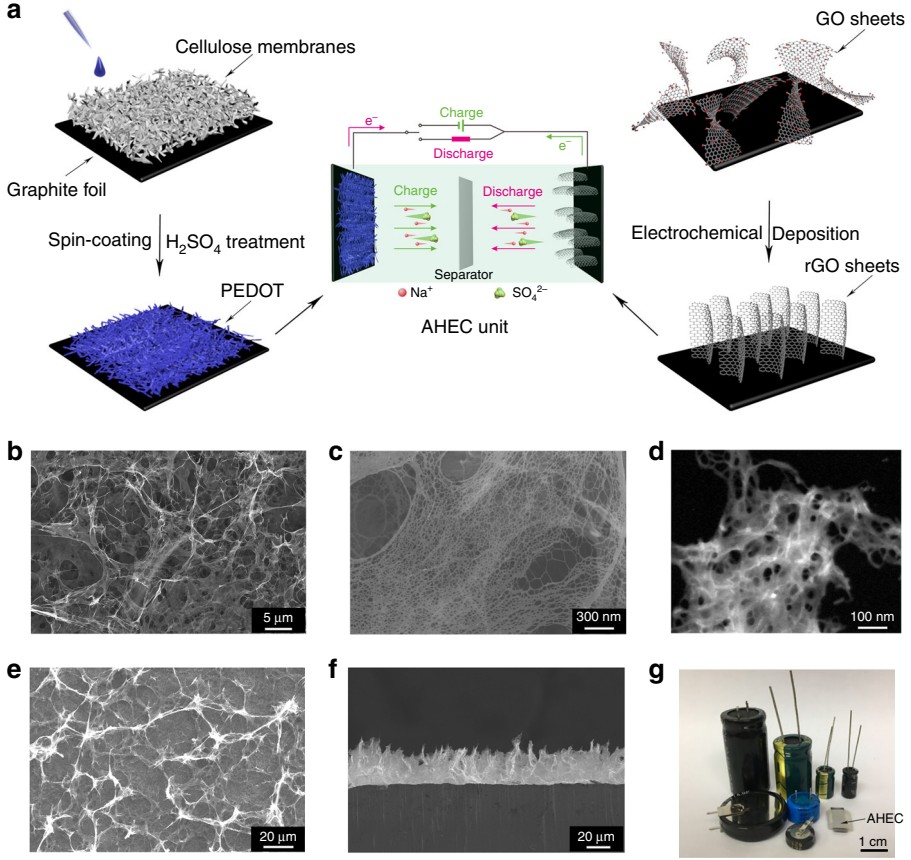

**Fig. 1** Preparation and architectures of electrode materials. **a** Schematic illustrations of the preparation of acid-pretreated poly(3,4-ethylenedioxythiophene) (PEDOT) positive electrode, electrochemically reduced graphene oxide (ErGO) negative electrode, and the assembly of aqueous hybrid electrochemical capacitor (AHEC), and GO represents graphene oxide. **b**, **c** SEM images of the PEDOT positive electrode with different magnification, respectively. **d** High-Angle Annular Dark Field TEM image of PEDOT positive electrode. **e**, **f** Top and cross-section SEM images of ErGO negative electrode, respectively. **g** Photo of the AHEC unit and various commercial aluminum electrolytic capacitors (AECs)

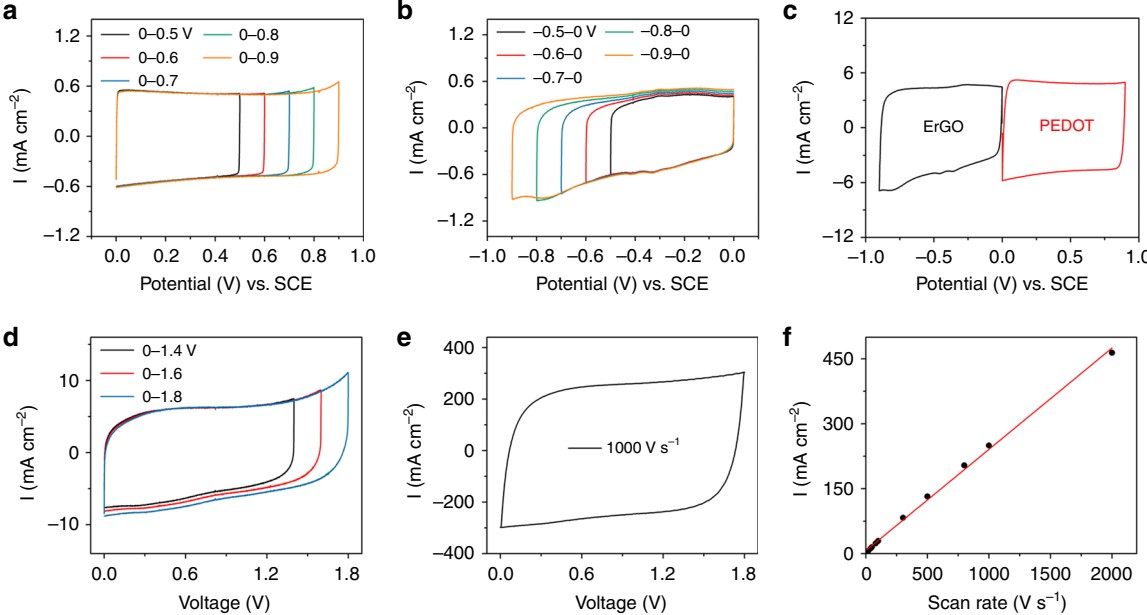

**Fig. 2** The cyclic voltammetry performances of single electrode and unit. **a**, **b** Cyclic voltammetry (CV) curves of the poly(3,4-ethylenedioxythiophene) (PEDOT) positive electrode (**a**) and electrochemically reduced graphene oxide (ErGO) negative electrode (**b**) in different potential ranges at scan rate of 1 V s$^{-1}$. **c** CV curves of PEDOT and ErGO electrodes in separate potential windows at scan rate of 10 V s$^{-1}$. **d** CV curves of the aqueous hybrid electrochemical capacitors (AHECs) in different voltage ranges at scan rate of 10 V s$^{-1}$. **e** CV curves of the AHEC at scan rate of 1000 V s$^{-1}$. **f** Plots of discharge current density versus scan rate of the AHEC. Source data are provided as a Source Data file

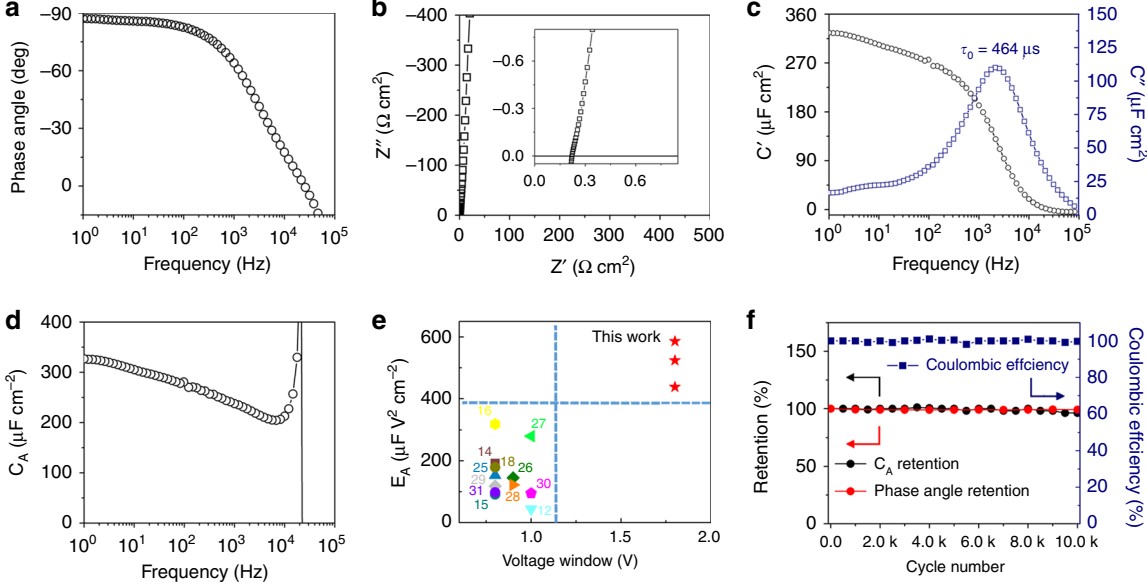

**Fig. 3** The electrochemical performances of unit. **a** Plots of phase angle versus frequency. **b** Nyquist plot; inset: the expanded view at high frequencies. **c** Plots of the real or imaginary part of specific capacitance ($C'$ or $C''$) versus frequency. **d** Plots of areal specific capacitance ($C_A$) as a function of frequency. **e** Comparison of $E_A$ and voltage window of the aqueous hybrid electrochemical capacitor (AHEC) with those of the reported aqueous AC-line filtering electrochemical capacitors (ECs) (Supplementary Table 1)[12,14–16,18,25–31]. **f** Cycling stability and the Coulombic efficiency of the AHECs at a current density of 5 mA cm$^{-2}$ for 10,000 cycles. Source data are provided as a Source Data file

(GCD) curves with a triangular shape also confirm the ideal capacitive behavior of the AHEC (Supplementary Fig. 11).

The good electrochemical performances of AHEC unit was further evaluated on electrochemical impedance spectroscopy (EIS). Generally, the phase angle at 120 Hz represents a "factor of merit" for AC response[24,25]. The AHEC shows a phase angle of −82° at 120 Hz (Fig. 3a), indicating the AHEC possesses the almost identical fast response capability compared with the commercial AEC (Supplementary Fig. 12a). At high-frequency region of Nyquist plots (Fig. 3b), the AHEC exhibits a low equivalent series resistance of 0.21 Ω cm$^2$. Also in the Nyquist plots (Fig. 3b), the $\tau_{RC}$ (time constant) of the AHECs (67 µF) is calculated to be as small as 0.18 ms, which is close to that (0.09 ms) of the AEC (22 µF). Considering the capacitance of AHEC is three times as much as that of AEC, the resistance of AHEC is reasonably lower than that of AEC (Supplementary Fig. 12b). Meanwhile, the $\tau_{RC}$ of the AHEC is better than most aqueous ECs and over 40 times shorter than the required period of 8.3 ms for 120 Hz filtering (Supplementary Table 1)[12,14–16,18,25–31]. Such a small $\tau_{RC}$ demonstrates the highly efficient reduction of charging/discharging time for fast AC-line filtering ability at 120 Hz[32].

The specific real ($C'$) and imaginary ($C''$) capacitances are plotted versus frequency (Fig. 3c). From the frequency ($f_o = 2155$ Hz) at maximum $C''$, the $\tau_0$ ($=1/f_o$) could be calculated to be 0.464 ms, also representing the fast ion adsorption/desorption process[24]. The $C_A$ of the AHECs is measured to be 270 µF cm$^{-2}$ at 120 Hz (Fig. 3d), which is much higher than that of bare graphite foils-based ECs (Supplementary Fig. 13), indicating the negligible influence of the substrate for the $C_A$. Furthermore, the capacitive performances of the AHEC could also be easily adjusted by increasing the electrode loading on the graphite foil without sacrificing its fast responsive capability (Supplementary Fig. 14). Combined with the voltage window of 1.8 V, the areal specific energy density ($E_A$) of the AHEC at 120 Hz was calculated to be 438 µF V$^2$ cm$^{-2}$, which is the highest among the reported aqueous ECs with a comparable phase angle (Supplementary Table 1; Fig. 3e). In addition, the AHEC also possesses a good electrochemical stability for practical

applications (Fig. 3f). Its capacitance and phase angle retentions are almost 100% after repeated charging/discharging at 5 mA cm$^{-2}$ for 10,000 cycles. Meanwhile, the Coulombic efficiency keeps 100% during this cycle test. Besides, the AHEC exhibits a relatively low leakage current of 3.5 µA (Supplementary Fig. 15), also representing its good stability.

**Filtering performances of unit.** According to the above electrochemical characterizations, the AHEC unit was further engineered into an experimental model rectifier and filter circuit to verify its filtering performance (Fig. 4a). The AHEC is capable to successfully smooth the AC into the DC (3.6 V$_{peak–peak}$, 60 Hz, Fig. 4b, Supplementary Movie 1) only with 20 mV fluctuation, which is 24 times less than that produced by AEC (293 mV) when the AHEC and AEC have similar Farad rating (AHEC: 67 µF, AEC: 22 µF). More impressively, along with the filtration of sinusoidal AC waveforms, AHEC can be well competent for arbitrary ripple filtering (Fig. 4c–i; Supplementary Fig. 16). Both the curved (e.g., square, triangle, Lorentz, decaying, stair, and electrocardiogram) and geometric (rhombus, ellipse, and heart) inputs can be rectified into the ideal linear outputs. Noteworthy, the AHEC owns the ability to efficiently filter noise with irregularly violent pulses, demonstrating the great potential and good filtering capability under versatile filtering conditions.

The frequency applicability is another standard for multifunctional and widely available AC-line filtering systems. This AHEC unit works well with a frequency range from 1–10,000 Hz (Fig. 5a; Supplementary Fig. 17). In such a broad frequency range, all the output signals maintain a smooth line shape with a negligible variance of <0.01, much smaller than those (almost 0.08) generated by the AEC (22 µF/ 450 V; Fig. 5b).

The filtering performance and wide frequency capability of this AHEC unit are mainly benefitted from its high specific capacitance and low total impedance. For the typical filtering process, a sinusoidal AC signal was firstly converted into one constant polarity signal after flowing through a bridge rectifier which is composed of four diodes (Fig. 5c, e; Supplementary

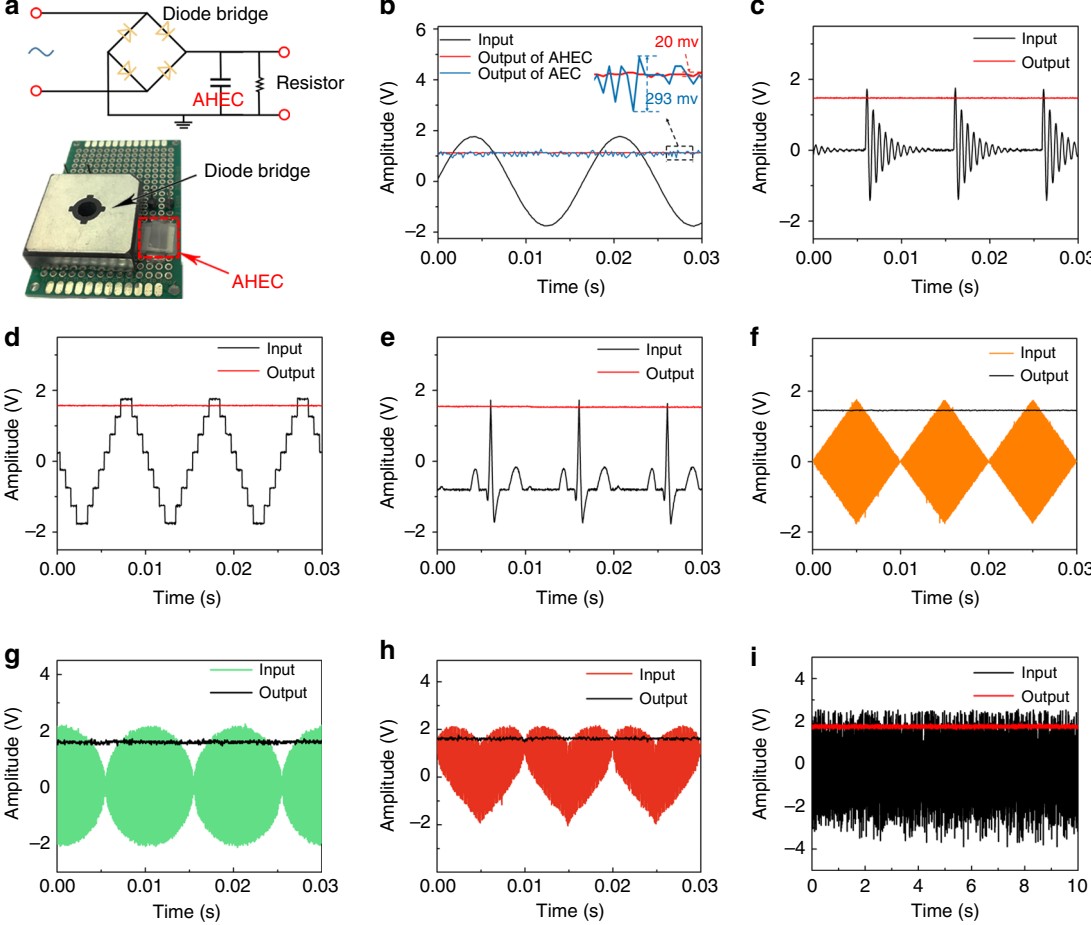

**Fig. 4** The filtering performances of unit. **a** Schematic: demonstration of this circuit for filtering circuit; optical image: the aqueous hybrid electrochemical capacitor (AHEC) and diode bridge on the board circuit for smoothing AC signals. **b** Comparison of AC line-filtering performances of the AHEC with the commercial AEC at 60 Hz. **c–i** The filtering performances of the AHEC unit for arbitrary waveforms. The input signals are decaying (**c**), stair (**d**), electrocardiogram (**e**), rhombus (**f**), ellipse (**g**), heart (**h**), and violent noise waveforms (**i**). The frequency of **c** to **h** is 100 Hz. Source data are provided as a Source Data file

Fig. 18). The output signal can be expressed as:

$$V = |V_0 \sin(\omega t)| \qquad (2)$$

where $V$ is the voltage signal, $V_0$ is the maximum voltage value, $\omega$ is angular frequency, $t$ is time. Subsequently, the rectifying signals were further smoothed by the filtering capacitor to produce a straight line with ripple voltage ($r$) (Fig. 5e). During this filtering process, the capacitor was continually charged and discharged like a reservoir. When the voltage rose above that of the capacitor, this capacitor charged up. And then as the voltage fell, the capacitor discharged to supply the stability of voltage (Fig. 5d). Thus, the $r$ was mainly derived from the voltage drop due to the discharging of capacitor. Furthermore, transient voltage ($V(t)$) of the capacitor could be described as:

$$V(t) = V_0 e^{\frac{-t}{RC}} \qquad (3)$$

Thus, $r$ could be calculated according to the equation (4)

$$r = V_0 - V(t) = V_0(1 - e^{\frac{-t}{RC}}) \qquad (4)$$

where $R$ is the resistance of the capacitor and $C$ is the capacitance of the capacitor.

According to Eq. (4), it can be concluded that $r$ has a positive correlation with $t/RC$. Furthermore, the ripple voltage will decay along with the increasing capacitance. Therefore, the large specific

energy density and ideal electric double-layer capacitive behavior of AHEC ensure its good filtering capability.

On the other hand, the low equivalent series resistance (ESR) and total impedance ($Z$) can efficiently reduce the internal temperature and power loss during operational state of capacitors, which are significant for reduction of the capacitors' ripple current and elongation of their working life[8]. Generally, for most capacitors, the typical equivalent circuits could be depicted as an RCL circuit (a resistor, a capacitor, and an inductor connected in series). Thus, the $Z$ has the following equations (5):

$$Z = \sqrt{\mathrm{ESR}^2 + \left(\frac{1}{2\pi f C} - 2\pi f L\right)^2} \qquad (5)$$

where $f$ is the frequency, $C$ is the capacitance, and $L$ is the inductance.

At the low-frequency region, the equivalent series inductance (ESL) could be neglected, and the low ESR and large $C$ are more favorable to lower the $Z$. Meanwhile, at the high-frequency region, the $Z$ is mainly influenced by the ESR and ESL of the capacitor[8]. As shown in Supplementary Fig. 19, the AHEC unit exhibits a less $Z$ within all the frequency range than that of AEC. Therefore, the AHEC unit demonstrates a great promise for AC/DC conversion within frequency range below the self-resonant frequency (Supplementary Fig. 19). Furthermore, in order to reach a higher workable frequency range, a strategy to adjust self-

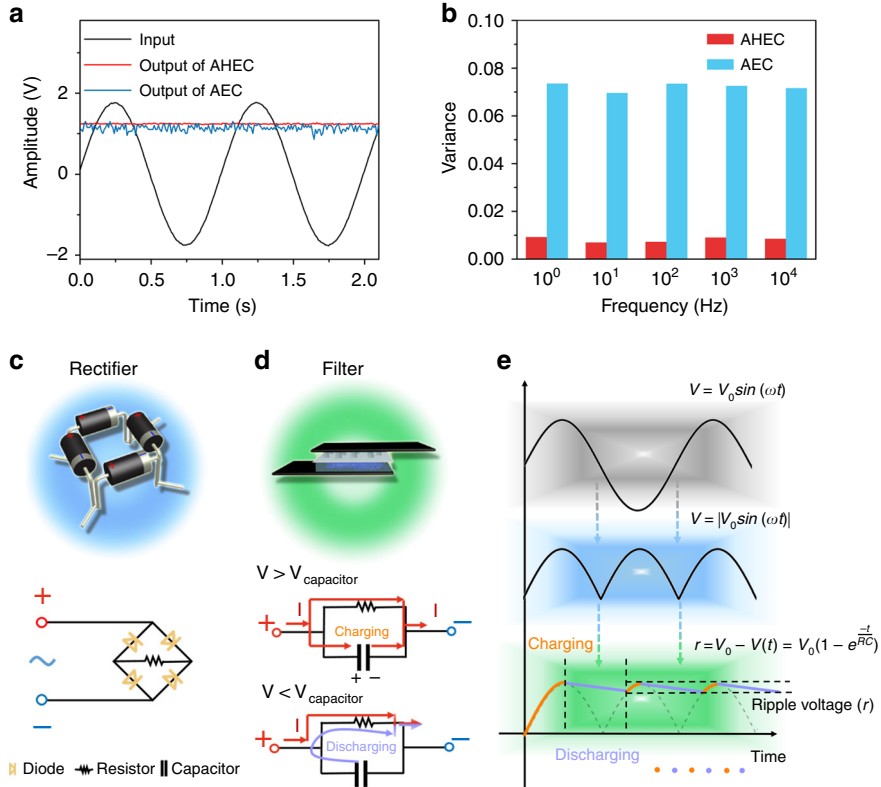

**Fig. 5** The frequency filtering performances and filtering process. **a** AC line-filtering performance of the aqueous hybrid electrochemical capacitor (AHEC) unit and aluminum electrolytic capacitors (AEC) at 1 Hz. **b** Histogram of variance of the output signals data of AHEC and AEC vs. frequency. **c** The schematic illustration of rectifier and its circuit. **d** The schematic illustration of AHEC unit and its filtering procedure. **e** The conversion process of the sinusoidal AC signal to DC signal. Source data are provided as a Source Data file

resonant frequency is changing capacitances of AHEC. When the capacitance of AHEC is about 1 μF, its self-resonant frequency is 500 kHz, demonstrating its potential for filtering within a wider frequency range (Supplementary Figs. 20 and 21).

**Filtering performances of units integration.** Easy integration engineering of AHEC units provides the versatile function for practical applications. As shown in Fig. 6a, an in-plane assembled AC-line filtering device can be easily obtained by connecting AHEC units in sequential manner on the plastic substrate. More specifically, the AHEC unit is connected by the exposed graphite foils which act as wires and are fixed by polyimide tapes. The obtained flexible device can be easily bent and curved without sacrificing its integrality (Fig. 6b and Supplementary Figs. 22 and 23). After connecting the units in series, a quasi-rectangular shape of the CV curves maintains up to 110 V (even to 200 V; Fig. 6c and Supplementary Fig. 24), which is the highest operating voltage reported so far to the best of our knowledge. Despite an increased resistance and a reduced capacitance of the integrated devices with increasing series amounts of AHEC unit (Supplementary Fig. 25), they maintain the fast frequency-response capability because of nearly unchanged time constant (RC) and phase angle retention at 120 Hz (Fig. 6d). Meanwhile, the integrated AHECs showed a higher $C_v$ than commercial AECs up to 16 V, and is comparable to commercial AECs within voltage range below 100 V in AC line filtering (Supplementary Fig. 26)[33]. The vertical Nyquist plots of the connected ECs suggests that each electrode is resistively coupled to its current collector and the porous electrode behavior is not obvious. And the nearly vertical line at low frequencies implies the pure capacitive behavior (Supplementary Fig. 25)[34,35].

Moreover, the planar AHEC integration can work at much high voltage (e.g., 220 $V_{peak-peak}$ and 400 $V_{peak-peak}$) within a wide frequency range (10–10,000 Hz, Supplementary Fig. 29). Those integrated units exhibit the same ability to filter various input waveforms into linear ones (Fig. 6e and Supplementary Figs. 27 and 28) with customizability of voltage. This report of filtering capacitors simultaneously achieves the relatively wide frequency range and high adapted voltage on demand without any sacrificing its filtering performance (Fig. 6f and Supplementary Table 2)[6,12,17–19,27,29,30,36–38]. Therefore, the AHECs have a great potential for the applications in line-powered electronics, a promising replacement of bulky AEC in line-filtering practically.

## Discussion

A unique AHEC was developed for AC-line filtering with high specific capacitance and enlarged voltage window, which work well with a wide frequency range from 1 Hz to 10,000 Hz and an accumulative voltage of hundreds of volts. Meanwhile, the AHEC displays a record high areal specific energy density of 438 μF $V^2$ cm$^{-2}$ at 120 Hz, and can filter arbitrary AC waveforms to smooth DC within wide frequency range and high-voltage condition. Considering the high-energy density and good filtering properties, the as-prepared AHEC presents a great promise for future multi-scenario and universally applicable filtering capacitors of practical importance.

## Methods

**Synthesis of graphene oxide**. GO were synthesized by a modified Hummers method[39]. Then the GO dispersion was treated by ultrasonication for 30 min using a SBL-15DT ultrasonic constant temperature cleaning instrument (Scientz, China). The SEM images, Attenuated total reflection Fourier transform infrared (FTIR) spectra, UV-vis spectra of GO sheets were showed in Supplementary Fig. 30.

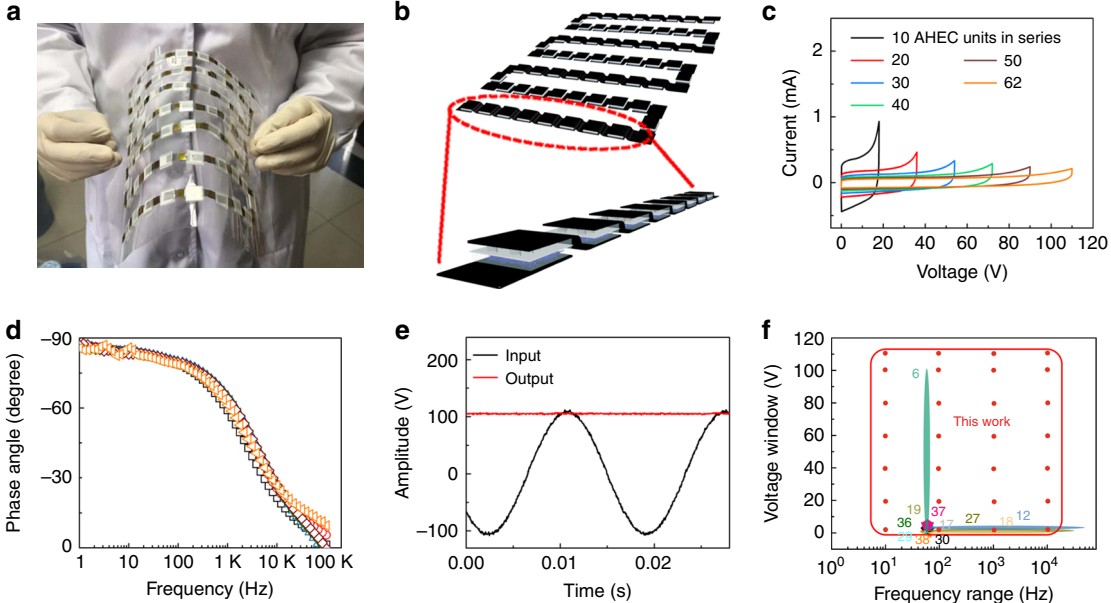

**Fig. 6** The electrochemical performances of integrated units. **a**, **b** Photograph (**a**) and schematic illustrations (**b**) of flexible 62 aqueous hybrid electrochemical capacitor (AHEC) units connected in series. **c** CV curves for different numbers of AHECs connected in series (scan rate: 10 V s$^{-1}$). **d** Plots of phase angle versus frequency of different numbers of AHECs connected in series. **e** AC line-filtering performances of 62 AHEC units connected in series. **f** Comparison of the voltage window and frequency demonstration of the AHECs with those of the reported AC-line filtering ECs, the red curves and circles represent the voltage and frequency range in this work (Supplementary Table 2)[6,12,17–19,27,29,30,36–38]. Source data are provided as a Source Data file

**Preparation of positive electrode**. A cellulose membrane (0.5 × 0.5 cm$^2$) was put a highly conductive and flexible graphite foil (thickness: 20 µm, conductivity: 15,000 s cm$^{-1}$, and tensile strength: 110 MPa, Supplementary Fig. 31). Then 10 µL dilute PH1000 (PEDOT:PSS) solution was dropped on a cellulose membrane (Supplementary Fig. 32), followed by a spin-coating treatment to ensure uniform and sufficient penetration of the PH1000 solution into cellulose membrane. Then, the wet membrane was immediately immersed in concentrated sulfuric acid for more than 24 h at room temperature. During this process the cellulose membrane was gradually dissolved by the acid, maintaining the integrity of the acid treated PEDOT film on the graphite foil. The as-prepared PEDOT coated graphite foil was cut into the shape as showed in Supplementary Fig. 1. These tailored electrodes were then cleaned with dilute sulfuric acid (12 M, 6 M, and 1 M) and deionized water step by step.

**Preparation of negative electrode**. The negative electrode materials were prepared by an electrochemical assisted decomposition method, which could simultaneously induce reduction and assembling of GO sheets into a 3D interpenetrating porous graphene structure along the direction of electric field[15,19,40]. In a typical procedure, 3 mg mL$^{-1}$ GO dispersions containing 0.1 M LiClO$_4$ electrolyte were prepared, and the GO was electrolyzed on the graphite foil at an applied potential of −1.2 V vs SCE for a given time by using a CHI760D potentiostat-galvanostat (CH Instruments Inc.). To be specific, graphite foils were used as the working electrodes, a gold foil was used as the counter electrode, and the potentials were referred to a SCE. Subsequently, the residual GO and ions were removed by repeated soaking in water and further deep electrochemical reduction in the 1 M LiClO$_4$ electrolyte at −1.2 V vs SCE for at least 30 s. Finally, the ErGO negative electrode was obtained by soaking it in water for several times to remove the residual salt ions within electrode.

**Electrochemical characterizations of single electrodes**. The PEDOT and ErGO were characterized by the three electrode system, respectively. A Pt mesh, an Ag/AgCl (saturated KCl) system, and 1M Na$_2$SO$_4$ serve as the counter electrode, reference electrode, and electrolyte, respectively.

**Fabrication of unit**. An AHEC was fabricated by assembling a PEDOT electrode and an ErGO electrode in a sandwiched manner, and was sealed with parafilm. Specifically, a PEDOT electrode (0.5×0.5 cm$^2$) and an ErGO electrode (0.5 × 0.5 cm$^2$) were used as the positive electrode and negative electrode for an AHEC unit with 1M Na$_2$SO$_4$ as the electrolyte. Porous anodic aluminum oxide membranes (Whatman, Anodisc 25) were employed as the separators. In case of the series and parallel strings, porous anodic aluminum oxide was substituted by cellulose membranes for better flexibility. The capacitive performance of AHEC could be adjusted by material loading of electrode. The different AHEC units were

nominated AHEC$_n$ ($n$ = 2, 4, and 6, representing the time (2 s, 4 s, and 6 s) of the electrochemical assisted decomposition of the ErGO at the applied potential of −1.2 V). For the positive electrode, the concentration of pristine PH1000 solution were 0.8 wt.% for AHEC$_2$, 1 wt.% for AHEC$_4$, and 1.2 wt.% for AHEC$_6$.

**Fabrication of high-voltage strings of units**. Different amounts of AHEC were connected in series or in parallel. More specifically, the exposed graphite foils of each electrode, acting as wires, were contacted with each other, and were fixed by polyimide tapes. And every AHEC was fixed on a transparent polyethylene glycol terephthalate film with double-sided tape.

**Electrochemical characterizations of two-terminal devices**. The AHEC was tested by using a CHI 660E Potentiostat (CH Instruments Inc., China) in a two-electrode system. For CV and GCD measurements, the electrochemical windows were controlled to be 0–1.8 V. EIS tests were carried on at 5 mV amplitude in the frequency range of ~10$^6$–1 Hz. For the CV test of connected AHEC in series or in parallel were carried out by using a Keithley 2612. In the AC-line filtering test, all the input signals were supplied by an 33511B arbitrary function generator (Agilent Technologies Inc., Tektronix, USA), and then the signals were performed with a GBPC3005W ready-made single-phase silicon bridge rectifier (Sep Electron. Corp., Taibei, China). For the AC-line filtering test of connected AHECs, the input signals were enlarged by an ATA-2041 high-voltage amplifier (Agitek, China). All the outputs were recorded by the use of a RTB2002 mixed domain oscilloscope (Rohde & Schwarz, Germany).

The specific areal capacitance ($C_A$, µF cm$^{-2}$) was calculated by:

$$C_A = -\frac{1}{2\pi f Z'' S} \quad (6)$$

The volumetric capacitance ($C_V$, F cm$^{-3}$) of the device was calculated by:

$$C_V = \frac{C_A}{d} \quad (7)$$

The resistor-capacitor time constant ($\tau_{RC}$) was calculated by:

$$\tau_{RC} = -\frac{Z'}{2\pi f Z''} \quad (8)$$

The real or imaginary specific areal capacitance ($C'$ or $C''$) were calculated by:

$$C' = -\frac{Z''}{2\pi f |Z|^2 S} \quad (9)$$

$$C'' = -\frac{Z'}{2\pi f |Z|^2 S} \quad (10)$$

The $\tau_0$ was derived from the frequency at maximum $C''$:

$$\tau_0 = \frac{1}{f_0} \tag{11}$$

The areal energy density ($E_A$) of was calculated by:

$$E_A = \frac{C_A U^2}{2} \tag{12}$$

In the equation above, $f$ is frequency; $d$ the thickness of the AHEC including the current collects, electrode materials, separator, and the uncompacted gaps; $Z'$ or $Z''$ is the real or imaginary impedance; $S$ is the area of electrode; $f_0$ is frequency at maximum $C''$.

**Characterization**. Scanning electron microscopy (SEM) micrographs were achieved by using a Sirion 200 field-emission SEM (FEI Corporation, USA). Transmission electron microscopy (TEM) images and EDS were obtained from the Tecnai F20 transmission electron microscope (FEI Corporation, USA). XPS spectra were recorded on using an Escalab 250 photoelectron spectrometer (ThermoFisher Scientific, USA). Raman spectra test were carried out with the aid of a LabRAM HR Evolution with a 532-nm laser (HORIBA Jobin Yvon, France). Mechanical tensile tests were conducted on an Instron 3342 universal testing machine (Instron, USA) with a constant loading rate of 0.5 mm min$^{-1}$ and a gauge length of 10 mm. Electrical conductivity was measured by a four-point technique via a four-point probe KDY-1 sheet resistivity tester (Kunde Technology, Guangzhou, China). The thickness of conductive film is measured by SEM images and a P-7 stylus profiler (KLA-Tencor, USA).

## Data availability
The data that support the findings of this study are available from the corresponding authors upon reasonable request.

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

## Acknowledgements
We thank Zishuai Li for the providing of 33511B arbitrary function generator. This work was supported by the financial support from the National Key R&D Program of China (2017YFB1104300, 2016YFA0200200), National Science Foundation of China (No. 51673026, 51433005, and 21674056), NSFC-MAECI (51861135202).

## Author contributions
M.W. performed the AHEC unit experiments; F.C. performed integrated AHECs experiments. M.W. and F.C. wrote the manuscript. H.G., H.M., M.Z., T.G., and C.L. gave the advice of experiments and reviewed manuscript. All authors discussed the results and revised this paper. L.Q. designed and supervised all the work.

## Additional information

**Competing interests:** The authors declare no competing interests.

**Peer Review Information:** *Nature Communications* would like to thank Shuai Wang and other, anonymous, reviewer(s) for their contribution to the peer review of this work. Peer review reports are available.

