## [Peer Review File · Nature Communications]

Reviewer #1 (Remarks to the Author):

This manuscript describes high-frequency electrochemical capacitor electrodes and two-terminal devices made from these electrodes including very high-voltage devices. Converting 100 Hz AC of various waveforms to DC is shown. The topics presented are of interest to the electrochemical capacitor community--electrode fabrication aspects and two-terminal device fabrication in series and parallel strings appears to be novel. However, insufficient details are included in this manuscript about electrode and device fabrication to make it acceptable for publication. A major revision is needed that includes electrode fabrication details well beyond the brief descriptions given on pages 17, 18. Fabrication details of high-voltage strings of AAS units, which were not described, are also needed. In addition, the revision should correct several errors, which includes:

1. The word "supercapacitor" used in the title and in the text of the manuscript is the name of a product and not the name of a technology. A preferred name would be "electrochemical capacitor", which is the name of a technology.
2. The word "asymmetric" used in the title and in the text of the manuscript is an incorrect descriptor. This word was first used in the literature in 2001 in US patent 6,222,723 titled "Asymmetric electrochemical capacitor and method of making", which provides the definition for this word. It is an energy storage device that has one electrode with at least three-times larger capacitance than the second electrode. Equation 1 on page 8 of the manuscript states that the two electrodes have equal capacitance. Thus, the word "asymmetric" is improperly used in the manuscript. This reviewer suggests the word "hybrid" to be the proper substitute word.
3. Page 1, line 18: The statement that the AAS is applicable for smoothing AC to DC over the frequency range 10 to 1,000,000 Hz is not supported by data given in the manuscript. Figure 3d shows that the self-resonant frequency of an AAS unit is ~20 kHz. This would be the upper operating frequency for it to act as a capacitor smoothing filter. Above this frequency, the AAS unit behaves like an inductor, not a capacitor.
4. page 2, lines 32-33: "satisfying the smooth of arbitrary waves inputted and the" should be revised so as to improve the English.
5. page 2, line 36: The "huge internal resistance" described is relative to what value of resistance and for what size device? A better development in the manuscript would be to use the concept of "response time", i.e. the RC product. The RC product is invariant with the size of the device and thus does not beg this question.
6. page 3, lines 53: It is better to use the words "positive electrode" and "negative electrode" since anode and cathode are meant the opposite for battery readers than for capacitor readers and thus will increase clarity.
7. page 6, line 95: Ion diffusion should be removed from the list because it plays absolutely no charge storage role in electrodes that store surface charge. On the other hand, porous electrode behavior could be added since this can be important.
8. page 7, line 117: The reference electrode should always be stated when a potential is listed. Figures 2a-2c list vs SCE and the text should also list the same.
9. page 10, line 179: The time needed to reach 10,000 cycles at 120 Hz is 83 seconds, which definitely is not a "long-term cycle" test as needed for AC to DC capacitive filtering technology. A more accurate statement would be to use "cycle test" instead of "long-term cycle".
10. page 11, Figure 4b: This comparison is useful only if capacitor Farad rating of the AAS and the

AEC are provided.

11. page 201, line 201: The word "superior" may be justified, but this is not known because the Farad rating of the AAS and the AEC are not given.

12. page 16, line 271-272: The electrochemical property of phase angle is independent of the series number and connection style. However, the electrochemical properties of "capacitance" and "series resistance" do depend on the series number and the connection style. Thus, the statement in the manuscript is not correct.

13. page 16, lines 277-279: The fact that there is no semicircle in the Nyquist plot simply means that each electrode is resistively coupled to its current collector. Nothing can be inferred about ion diffusion speed from this behavior. The fact that there is no 45 degree intersection with the real axis in the Nyquist plot simply means that there is no porous electrode behavior. Nothing can be inferred about ion diffusion speed from this behavior.

14. page 17, line 292: The statement "which works well" was not demonstrated. The distribution of voltages must be show with no units at excessive voltage to make this claim. To make such a claim, this reviewer suggests showing the distribution of unit voltages and the maximum unit voltage on those in series.

15. page 17, line 297: "Safety" was not addressed in the body of the manuscript. Thus the statement made in the conclusion is not supported and should be removed. Alternatively, safety could be discussed but this is not recommended since this would be a separate new topic that would increase the length of the manuscript by many pages for proper treatment plus, safety is not the subject of this study.

16. It would be of great interest to show a comparison with prior results, especially those reported in Figure 10 of the Journal of the Electrochemical Society, 162, A5077-A5082 (2015). This comparison may help to demonstrate the significance of the current work.

Reviewer #2 (Remarks to the Author):

The transform of current with disorder volts to direct current is of great importance in the fields of power generation, especially friction power generation, wind power generation, also steam response power generation reported by parts of the authors. Wu et al. reported an asymmetric aqueous supercapacitor for AC-line filtering with high specific capacitance and enlarged voltage window, which is of particular importance to work well with an ultra-wide frequency range from 10 Hz to 1,000,000 Hz and an accumulative voltage of hundreds of volts. Specially, it can filter arbitrary AC waveforms to smooth DC within ultrawide frequency range and high voltage condition. This work indeed presents a great advance for multi-scenario and universally applicable filtering capacitors for practical applications. Therefore, I would recommend the acceptance after addressing the following minor concerns.

1. The reason to choose PEDOT as cathode and graphene as anode should be discussed in the manuscript.
2. Figure 1f shows the oriented structure of the anode. It is better to point out why graphene exhibited such structure.
3. When discussing the resistance of AAS or AEC, the authors should provide the corresponding results.
4. "The excellent filtering performance and wide frequency capability of this AAS unit are mainly benefited from its high specific capacitance and low total impedance." (Page 12, line1-2). Please give a brief description of why the supercapacitor exhibited high specific capacitance and low total

impedance, and how the anode and the cathode affect these performances.

5. For practical application, it is important to evaluate the leakage current and self-discharge characteristics of the supercapacitor devices, please supply these data.

6. As the author demonstrate a flexible integrated AASs, how is the flexibility and mechanical stability of the devices on working state.

7. There is an AEC used for comparison in this work. Please give detailed technical parameters of it.

8. For practical application, the author can fabricate solid-state and/or organic AASs?

9. Language and description should be rechecked to make sure correction display. For example, in Page 3, line 3, it is better to call this filter rather than filtering. Page 3, line 15, meet is better than match. Additionally, please rearrange the following sentence, because it looks like uncompleted. Page 6, line 7, "be" should be deleted.

Reviewer #3 (Remarks to the Author):

The authors reported an aqueous asymmetric supercapacitor exhibited excellent electrochemical performances. The cathode and anode synergistically help the supercapacitor realize the efficient conversion of arbitrary AC to DC under an ultra-wide frequency range with hundreds of volts, promising for the renewable power generators. It is a very interesting and important work toward the ultrafast supercapacitors, and greatly broaden the potential application of line-filter capacitors fields. The characterization is also praiseworthy. For these reasons, I believe that this work shall attract wide attention and therefore the manuscript can be accepted for publication in Nature communication after minor revision.

1. In Page 13, line 20, what is the "its" mean?

2. Please provide the parameters and details of the raw materials including GO and PEDOT.

3. The electrical conductivity is important for AC filtering. How is the electrical conductivity measured? Please describe the details in the experimental section.

4. For the preparation process, whether the ErGO electrode was washed in water after deep further deep reduction? Any influence?

5. The ultrawide frequency is valuable for practical applications. The frequency range should be checked. For example, the AAS is developed for with an applicable ultra-wide frequency range from 1 to 1,000,000 Hz (Fig. 5a). But authors reported "from 10 Hz to 1,000,000 Hz. Furthermore" (Page 3, line 13).

6. The properties and functions of PEDOT and graphene acting as electrode should be briefly discussed in introduction.

7. It is better to improve the use of English language throughout the manuscript. For example, in Page 12 line 3, "a sinusoidal AC signal was firstly converted into one constant polarity signal by flowing through a bridge rectifier..."; in Page 13, line 3 "the rectifying signals was further smoothed..."

Point-by-Point Responses to the Reviewer's Comments

Reviewer #1 (Remarks to the Author):

This manuscript describes high-frequency electrochemical capacitor electrodes and two-terminal devices made from these electrodes including very high – voltage devices. Converting 100 Hz AC of various waveforms to DC is shown. The topics presented are of interest to the electrochemical capacitor community--electrode fabrication aspects and two-terminal device fabrication in series and parallel strings appears to be novel. However, insufficient details are included in this manuscript about electrode and device fabrication to make it acceptable for publication. A major revision is needed that includes electrode fabrication details well beyond the brief descriptions given on pages 17, 18. Fabrication details of high-voltage strings of AAS units, which were not described, are also needed. In addition, the revision should correct several errors, which includes:

Reply: Thank you so much for your positive recommendation. We also appreciate your careful reading of our manuscript and constructive suggestions for correcting and improving our manuscript. As suggested, we further described the preparation procedure of the electrode and the high-voltage strings. For your convenience, this information are listed below:

Preparation of PEDOT positive electrode. A cellulose membrane ($0.5 \times 0.5 \text{ cm}^2$) was put a highly conductive and flexible graphite foil (thickness: $20 \text{ }\mu\text{m}$, conductivity: $15,000 \text{ S cm}^{-1}$, and tensile strength: 110 MPa , Fig. S30). Then $10 \text{ }\mu\text{L}$ dilute PH1000 (PEDOT:PSS) solution was dropped on a cellulose membrane (Fig. S31), followed by a spin-coating treatment to ensure uniform and sufficient penetration of the PH1000 solution into cellulose membrane. Then, the wet membrane was immediately immersed in concentrated sulfuric acid for more than 24 hours at room temperature. During this process the cellulose membrane was gradually dissolved by the acid, maintaining the integrity of the acid treated PEDOT film on the graphite foil. The as prepared PEDOT coated graphite foil was cut into the shape as showed in Fig. S1. These tailored electrodes were then cleaned with dilute sulfuric acid (12 M, 6 M, and 1 M) and deionic water step by step. **(Please see Method part page 17, line 320–331.)**

Preparation of ErGO negative electrode. The negative electrode materials were prepared by an electrochemical assisted decomposition method, which could simultaneously induce reduction and assembling of GO sheets into a 3D interpenetrating porous graphene structure along the direction of electric field. In a

typical procedure, 3 mg mL⁻¹ GO dispersions containing 0.1 M LiClO₄ electrolyte were prepared, and the GO was electrolyzed on the graphite foil at an applied potential of -1.2 V vs SCE for a given time by using a CHI760D potentiostat-galvanostat (CH Instruments Inc.). To be specific, graphite foils were used as the working electrodes, a gold foil was used as the counter electrode, and the potentials were referred to a SCE. Subsequently, the residual GO and ions were removed by repeated soaking in water and further deep electrochemical reduction in the 1 M LiClO₄ electrolyte at -1.2 V vs SCE for at least 30 s. Finally, the ErGO negative electrode was obtained by drowning it in water for several times to remove the residual salt ions within electrode. **(Please see Method part page 17–18, line 332–344.)**

Fabrication of AHEC unit. An AHEC was fabricated by assembling a PEDOT electrode and an ErGO electrode in a sandwiched manner, and was sealed with parafilms. Specifically, a PEDOT electrode (0.5 × 0.5 cm²) and an ErGO electrode (0.5 × 0.5 cm²) were used as the positive electrode and negative electrode for an AHEC unit with 1M Na₂SO₄ as the electrolyte. Porous anodic aluminum oxide membranes (Whatman, Anodisc 25) were employed as the separators. In case of the series and parallel strings, porous anodic aluminum oxide was substituted by cellulose membranes for better flexibility. The capacitive performance of AHEC could be adjusted by material loading of electrode. The different AHEC units were nominated AHEC_n (n = 2, 4, and 6, representing the time (2 s, 4 s, and 6 s) of the electrochemical assisted decomposition of the ErGO at the applied potential of -1.2 V). For the positive electrode, the concentration of pristine PH1000 solution were 0.8 wt.% for AHEC₂, 1 wt.% for AHEC₄, and 1.2 wt.% for AHEC₆. **(Please see Method part page 18, line 349–361.)**

Fabrication of high-voltage strings of AHEC units. Different amounts of AHEC were connected in series or in parallel. More specifically, the exposed graphite foils of each electrode, acting as wires, were contacted with each other, and were fixed by polyimide tapes. And every AHEC was fixed on a transparent polyethylene glycol terephthalate film with double-sided tape. **(Please see Method part page 18–19, line 362–366.)**

1. The word “supercapacitor” used in the title and in the text of the manuscript is the name of a product and not the name of a technology. A preferred name would be “electrochemical capacitor”, which is the name of a technology.

Reply: We are grateful for this comment. The “supercapacitor (S)” have changed to the “electrochemical capacitor (EC)” in the revised manuscript. Meanwhile, according to your suggestion in comment 2, we also replaced the “asymmetric” with

“hybrid”. Thus, the abbreviation of “AAS” was modified to “AHEC”. (Throughout the manuscript)

2. The word “asymmetric” used in the title and in the text of the manuscript is an incorrect descriptor. This word was first used in the literature in 2001 in US patent 6,222,723 titled “Asymmetric electrochemical capacitor and method of making”, which provides the definition for this word. It is an energy storage device that has one electrode with at least three-times larger capacitance than the second electrode. Equation 1 on page 8 of the manuscript states that the two electrodes have equal capacitance. Thus, the word “asymmetric” is improperly used in the manuscript. This reviewer suggests the word “hybrid” to be the proper substitute word.

Reply: We agree with your suggestion, and have modified the word “asymmetric” to “hybrid”. The revised parts were marked by red in revised manuscript. (Throughout the manuscript)

3. Page 1, line 18: The statement that the AAS is applicable for smoothing AC to DC over the frequency range 10 to 1,000,000 Hz is not supported by data given in the manuscript. Figure 3d shows that the self-resonant frequency of an AAS unit is ~20 kHz. This would be the upper operating frequency for it to act as a capacitor smoothing filter. Above this frequency, the AAS unit behaves like an inductor, not a capacitor.

Reply: We agree with your instructive comment. According to the equivalent circuit (Fig. R1), a real capacitor should include equivalent series resistance (ESR), and equivalent series inductance (ESL). The self-resonant frequency (SRF) is the frequency at which the capacitive impedance ($\frac{1}{2\pi fC}$) equals the inductive impedance ($2\pi fL$). Above the SRF, the inductive behavior may limit the ability to shunt ripple due to the increasing inductive impedance [Paul *IEEE T. Ind. Appl.* **1992**, 34, 130–133; Neugebauer et al. *IEEE T. Ind. Appl.* **2004**, 40, 483–491]. While in real electrical circuits, a capacitor cannot be an inductor completely even at high frequency because of its inherent capacitor structure. And inductive impedance can only suppress its ability to build an electrical field but does not completely eliminate them. Thus, the capacitors in our manuscript can achieve AC line filtering possibly due to the constructed electrical field. Also, we further confirmed that in the Fig R2. Both the AHEC (~60 μ F at 120 Hz) and AEC (~60 μ F at 120 Hz) could be charged up at high frequency of 1,000,000 Hz, demonstrating their filtering capability at high frequency. At the same time, above the SRF, the filtering capability of capacitor is indeed weakened because of the arisen inductive impedance (Fig. 5b). In order to clarify this point, we added the discussion

about the SRF in the revised manuscript: “In addition, it should be mentioned, when the frequency is higher than the self-resonant frequency (SFR) at which the capacitive impedance ($\frac{1}{2\pi fC}$) equals the inductive impedance($2\pi fL$), the capacitor (including the AHEC and AEC) can also achieve the ripple filtering. That may be because the electrical field of the capacitors can be constructed even though the arisen inductive behavior would limit them (Fig. S20)”. (Please see page 14, line 261–266 and Supplementary Information, page 23 Fig. S20.)

Fig. R1. The equivalent circuit of a real capacitor.

Fig. R2. The initial stage AC filtering curves of AHEC and AEC at 1,000,000 Hz.

4. page 2, lines 32-33: “satisfying the smooth of arbitrary waves inputted and the ...” should be revised so as to the improve the English.

Reply: As you suggested, we have modified the sentence to “satisfying the smooth of arbitrary waves and...”, and then we further double checked and corrected the manuscript including improper expressions and typing errors. (Please see page 2, line 32–33.)

5. page 2, line 36: The “huge internal resistance” described is relative to what value of resistance and for what size device? A better development in the manuscript would be to use the concept of “response time”, i.e. the RC product. The RC product is invariant with the size of the device and thus does not beg this question.

Reply: Per your suggestion, we have revised the discussion about the resistance. Meanwhile, RC time is indeed a critical parameter to evaluate the electrochemical capacitor. The RC time is measured by the internal capacitance and resistance of the

capacitor. During the charging process, a low RC time is necessary for the fast charging at the high frequency. On the other hand, during the discharging process, a large RC time would give a slow decrease of the voltage, leading to a relatively small ripple voltage. Therefore, the RC time, as an essential character, could be used as an assessment for the capacitor. The statements were modified to “However, conventional filtering capacitors such as aluminum electrolytic capacitors (AECs) fail to meet the diversified demands because of its low specific capacitances that narrows its working frequency range.” (Please see page 2, line 33–36.) and “Also in the Nyquist plots (Fig. 3b), the τ_{RC} (time constant) of the AHECs (67 μF) is calculated to be as small as 0.18 ms, which is close to that (0.09 ms) of the AEC (22 μF). Considering the capacitance of AHEC is three times as much as that of AEC, the resistance of AHEC is reasonably lower than that of AEC (Fig. S12b).” (Please see page 8, line 161–164.)

6. page 3, lines 53: It is better to use the words “positive electrode” and “negative electrode” since anode and cathode are mean the opposite for battery readers than for capacitor readers and thus will increase clarity.

Reply: We agree with your opinion, and have substituted the words “positive electrode” and “negative electrode” for “cathode” and “anode”. (Throughout the manuscript)

7. page 6, line 95: Ion diffusion should be removed from the list because it plays absolutely no charge storage role in electrodes that store surface charge. On the other hand, porous electrode behavior could be added since this can be important.

Reply: According to your suggestion, the relative description had been deleted because there is indeed no ion transport process under this condition. Meanwhile, according to the porous electrode model [De Levie, *Electrochim. Acta.* **1963**, 8, 751–780; De Levie, *Electrochim. Acta.* **1964**, 9, 1231–1245], the equivalent circuit of a porous electrode with uniformly distributed electrolyte resistance and capacitance could be recognized as a transmission-line circuit when the pores were ideally assumed as cylindrical channels, which results in a 45° line in their Nyquist plots. Remarkably, a flat electrode with double layer capacitance would give a vertical line. In our case, a nearly vertical line to the real axis in Nyquist plots indicates that the porous electrode behavior is not obvious. Therefore, we have revised the statements to “The vertical Nyquist plots of the connected SCs suggests that each electrode is resistively coupled to its current collector and the porous electrode behavior is not obvious.” (Please see page 16, line 290–292.)

8. page 7, line 117: The reference electrode should always be stated when a potential is listed. Figures 2a-2c list vs SCE and the text should also list the same.

Reply: We have added the “vs SCE” where the reference electrode was used in the manuscript. **(Throughout the manuscript)**

9. page 10, line 179: The time needed to reach 10,000 cycles at 120 Hz is 83 seconds, which definitely is not a “long-term cycle” test as needed for AC to DC capacitive filtering technology. A more accurate statement would be to use “cycle test” instead of “long-term cycle”.

Reply: We agree with your point. And we have modified the original statement with the words “cycle test”. **(Please see page 7, line 124, and page 9, line 182.)**

10. page 11, Figure 4b: This comparison is useful only if capacitor Farad rating of the AAS and the AEC are provided.

Reply: Reply: According to your suggestion, we added the Farad rating at 120 Hz of AHEC (67 μF) and AEC (22 μF) in the revised manuscript, which is highlighted by red. **(Please see page 10, line 202.)**

11. page 201, line 201: The word “superior” may be justified, but this is not known because the Farad rating of the AAS and the AEC are not given.

Reply: We added the Farad rating of the AHEC (67 μF) and AEC (22 μF) in the revised manuscript. According to your suggestion, we further edited the description including “superior” with more appropriate word.

12. page 16, line 271-272: The electrochemical property of phase angle is independent of the series number and connection style. However, the electrochemical properties of “capacitance” and “series resistance” do depend on the series number and the connection style. Thus, the statement in the manuscript is not correct.

Reply: Yes. The capacitances and resistances are relative to the series amounts of the electrochemical capacitor. The equations are listed as follow,

$$\frac{1}{C} = \sum_{i=1}^n \frac{1}{C_i} \quad (R1)$$

$$R = \sum_{i=1}^n R_i \quad (R2)$$

where C is the total capacitance, C_i is the capacitance of the n^{th} capacitor, R is total resistance, and R_i is the resistance of the n^{th} capacitor.

Thus, the relative statement changed to “After connecting the units in series, a quasi-rectangular shape of the CV curves maintains up to 110 V (even to 200 V) (Fig. 6c; Fig. S23), which is the highest operating voltage reported so far. Despite an increased resistance and a reduced capacitance of the integrated devices with increasing series amounts of AHEC unit (Fig. R3), they maintains the fast frequency response capability because of nearly unchanged time constant (RC) and phase angle retention at 120 Hz (Fig. 6d).” (Please see page 15–16, line 282–288.)

Fig. R3 (a) Nyquist plots of the AHECs connected in series of different numbers. (b) Capacitances of the AHECs connected in series of different numbers. (Please see Supplementary Information page 27, Fig. S24.)

13. page 16, lines 277-279: The fact that there is no semicircle in the Nyquist plot simply means that each electrode is resistively coupled to its current collector. Nothing can be inferred about ion diffusion speed from this behavior. The fact that there is no 45 degree intersection with the real axis in the Nyquist plot simply means that there is no porous electrode behavior. Nothing can be inferred about ion diffusion speed from this behavior.

Reply: Thanks for this suggestion. According to the porous electrode model [De Levie, *Electrochim. Acta.* **1963**, 8, 751–780; De Levie, *Electrochim. Acta.* **1964**, 9, 1231–1245], the equivalent circuit of a porous electrode with uniformly distributed electrolyte resistance and capacitance could be recognized as a transmission-line circuit when the pores were ideally assumed as cylindrical channels, which results in a 45° line in their Nyquist plots. Remarkably, a flat electrode with double layer capacitance would give a vertical line. In our case, a nearly vertical line to the real axis in Nyquist plots indicates that the porous electrode behavior is not obvious. Therefore, we have revised

the statements to “The vertical Nyquist plots of the connected SCs suggests that each electrode is resistively coupled to its current collector and the porous electrode behavior is not obvious.” (Please see page 16, line 290–292.)

14. page 17, line 292: The statement “which works well” was not demonstrated. The distribution of voltages must be show with no units at excessive voltage to make this claim. To make such a claim, this reviewer suggests showing the distribution of unit voltages and the maximum unit voltage on those in series.

Reply: We are grateful for the insightful comments. According to your suggestion, the distribution of voltages was further displayed in the revised manuscript. To verify the actual working state, we connected 69 AHEC units in series and applied a constant voltage of 110 V. The specific distribution of unit voltages was measured by a multi-meter, as shown in Fig. R4. The maximum unit voltage is 1.86 V. Noteworthy, most of the units show a voltage below 1.8 V. The average voltage of the unit is $1.59 \text{ V} \pm 0.27 \text{ V}$. These data suggested that the series circuits could function well within a rational voltage range.

Fig. R4 The distribution of unit voltage in the series circuits (69 AHEC units and 110 V total voltage).

15. page 17, line 297: “Safety” was not addressed in the body of the manuscript. Thus the statement made in the conclusion is not supported and should be removed. Alternatively, safety could be discussed but this is not recommended since this would be a separate new topic that would increase the length of the manuscript by many pages for proper treatment plus, safety is not the subject of this study.

Reply: We have accordingly deleted the word “safety”. (Throughout the manuscript)

16. It would be of great interest to show a comparison with prior results, especially those reported in Figure 10 of the Journal of the Electrochemical Society, 162, A5077-A5082 (2015). This comparison may help to demonstrate the significance of the current work.

Reply: Thanks greatly for providing this valuable literature. The work [Miller et al. *J. Electrochem. Soc.* **2015**, 162, A5077–A5082] reported an efficient filtering capacitors based on carbon black deposited vertically-oriented graphene nanosheet (VOGN) electrode materials, which showed a high capacitance of 1.2 mF at 120 Hz after 5 minutes electro spray of carbon black. More importantly, they put forward an excellent concept of interconnecting planar multi-cell design which achieved volumetric advantages for the voltages below 100 V compared with commercial AECs. As suggested, we also compared the volumetric capacitance (C_v) variation of integrated AHECs to AECs, and added the related information in Figure S24. As shown in Fig. R5a, the thickness of an AHEC unit is about 60 μm , which is observed by the cross-section SEM images including two pieces of current collectors, positive electrode materials, negative electrode materials, and a piece of separator. With the C_A of 362 $\mu\text{F cm}^{-2}$ for the AHEC₆, the C_v of one unit is calculated to 0.06 F cm^{-3} . Assuming the C_v variation of integrated AHECs is only relative to the amounts of the AHEC units without any other loss. The curve of device capacitance density as function of the device voltage for the integrated AHECs could be depicted in Fig. R5b. The data of the commercial AECs were collected by previous literature and local stores [Miller et al. *J. Electrochem. Soc.* **2015**, 162, A5077–A5082]. The integrated AHECs show a higher C_v than AECs up to 16 V, and is comparable to AECs up to 100 V. **(Please see page 15, line 282–284.)**

In addition, we would like to cite the helpful reference in our manuscript. **(Please see page 22, line 474–475.)**

Fig. R5. (a) The thickness of an AHEC unit including current collector, electrode materials, and separator. (b) The volumetric capacitance variation as function of the voltages of the integrated AHECs and AECs. The data of the AECs are collected from

the previous literature [Miller et al. *J. Electrochem. Soc.* **2015**, 162, A5077–A5082] and local stores. (Please see **Supplementary Information, page 28 Fig. S24.**)

Reviewer #2 (Remarks to the Author):

The transform of current with disorder volts to direct current is of great importance in the fields of power generation, especially friction power generation, wind power generation, also steam response power generation reported by parts of the authors. Wu et al. reported an asymmetric aqueous supercapacitor for AC-line filtering with high specific capacitance and enlarged voltage window, which is of particular importance to work well with an ultra-wide frequency range from 10 Hz to 1,000,000 Hz and an accumulative voltage of hundreds of volts. Specially, it can filter arbitrary AC waveforms to smooth DC within ultrawide frequency range and high voltage condition. This work indeed presents a great advance for multi-scenario and universally applicable filtering capacitors for practical applications. Therefore, I would recommend the acceptance after addressing the following minor concerns.

Reply: We thank greatly for thinking our work important. To fully respond these comments, we conducted additional experiments and added more discussions in the revised manuscript. Some modifications and explanations are summarized as below.

1. The reason to choose PEDOT as cathode and graphene as anode should be discussed in the manuscript.

Reply: We chose the PEDOT and graphene as the electrodes mainly because of two factors: (1) their good physicochemical properties; (2) stable electrochemical stability. Generally, to ensure the fast responses capability (to 10 kHz or higher frequencies) of electrochemical capacitor, the electrode materials must be highly conductive, and the contact resistance between the electrode and current collector also need to be minimized [Fan. et al. *Nano Energy* **2017**, 39, 306–320]. On this basis, both the PEDOT and graphene electrodes have the excellent electrical conductivity [Xia et al. *Adv. Mater.* **2017**, 24, 2436–2440; Sheng et al. *Sci. Rep.* **2012**, 2, 247; Chi et al. *Adv. Energy Mater.* **2017**, 7, 1700591]. Meanwhile, they are well adhered on the highly conductive graphite foils ($\sim 15000 \text{ S cm}^{-1}$) in the form of a thin layer as shown in the Fig. S1,S5, also ensuring a minimum contact resistance. On the other hand, as shown in the SEM images (Fig. 1), the PEDOT and ErGO electrodes possess three-dimension network and large pores structure, which can not only provide the large surface for the capacitance, but also avoid the resistance from micropores or dead pores. Thus, the PEDOT and ErGO electrodes are ideal materials for constructing the high-frequency response electrochemical capacitor. Apart from that, the stability of the electrode is another important factor to be considered. We have characterized the voltage window of the two materials in the manuscript (Fig. S7 and S8). And they show the stable electrochemical behavior with PEDOT as the positive electrode (0-0.9 V vs SCE) and

ErGO as the negative electrode ($-0.9-0$ V vs SCE). Thus, the PEDOT and ErGO were chosen as the electrode materials. We added the brief descriptions about electrode materials in the Introduction part, which was marked in red. (Please see page 3, line 54–57.)

2. Figure 1f shows the oriented structure of the anode. It is better to point out why graphene exhibited such structure.

Reply: The orientation of graphene sheets was induced by the directional electric field during electrochemical deposition (Electrorheological behavior of GO sheets). Driven by directional electric field, GO sheets self-assembled onto the surface of substrate electrode along the direction of electric field during the electrodeposition process [Yu et al. *Analyst*, **2014**, 139, 4525–4531]. Simultaneously, they were electrochemically reduced to ErGO sheets to form a porous network because of π - π stacking and hydrophobic interactions.

3. When discussing the resistance of AAS or AEC, the authors should provide the corresponding results.

Reply: The resistance comparison between AHEC and AEC has been pictured in the Fig. S12b and S19, including the ESR and total impedance comparison. However, there is no detailed description next to the figures. For your convenience, we added some sentences to describe specific resistance comparison in the Supporting Information.

(1) “It is informative to compare the electrochemical performances and resistance of AHEC ($67 \mu\text{F}$) and AEC ($22 \mu\text{F}$). As shown in Fig. S12a, the AHEC has the comparable phase angle (over 80°) to the AEC at 120 Hz, indicating the fast response of the AHEC at high frequency condition. In addition, the AHEC exhibits a smaller ESR ($0.21 \Omega \text{ cm}^2$) compared with that ($4.1 \Omega \text{ cm}^2$) of AEC. This could also be confirmed by the RC time. The AEC possess a smaller RC time of 0.09 ms at 120 Hz than that (0.18 ms) of AHEC. But the capacitance of AHEC is three times higher than that of AEC. Thus, the resistance of the AEC is at least six times than that of AHEC.” (Please see Supplementary Information, page 15.)

(2) “Fig. S19 shows the impedance of AEC and AHEC within the frequency range from 1 Hz to 100,000 Hz. Both the AEC and AHEC exhibits a typical properties of RCL circuit including the capacitive impedance ($1/2\pi fC$), inductive impedance ($2\pi fL$), and ESR. The ESR could be read at the frequency that $1/2\pi fC = 2\pi fL$. It is also the lowest point in the curves. And, the AHEC has the smaller ESR ($0.21 \Omega \text{ cm}^2$). Meanwhile, the lower capacitance of AEC leads to the large capacitive impedance within the wide

frequency. Thus, the AHEC exhibits the low total impedance within the frequency range”
(Please see Supplementary Information, page 22.)

4. “The excellent filtering performance and wide frequency capability of this AAS unit are mainly benefited from its high specific capacitance and low total impedance.” (Page 12, line1-2). Please give a brief description of why the supercapacitor exhibited high specific capacitance and low total impedance, and how the anode and the cathode affect these performances.

Reply: As we mentioned in the reply of comment 1, the structures and properties of electrode materials would determine the electrochemical performances. For the PEDOT electrode and ErGO electrode, they both have porous three-dimension structure which can provide the abundant surface area for the active capacitive behavior. This view that large specific surface area would result in large capacitance has been accepted by many reports (Kötz et al. *Electrochim. Acta* **2000** 45, 2483–2498; Shao et al. *Chem. Soc. Rev.* **2015**, 44, 3639–3665 and so on). On the other hand, the excellent conductivity of these electrodes and well interface contact as well as high ion conductivity of the aqueous electrolyte guarantee the low ESR of $0.21 \Omega \text{ cm}^2$ in the electrochemical capacitor. Meanwhile, the high capacitance also lead to a relatively low capacitive impedance ($\frac{1}{2\pi fC}$) according to the equation (5). Thus, our electrochemical capacitors show a good total impedance within the wide frequency range.

5. For practical application, it is important to evaluate the leakage current and self-discharge characteristics of the supercapacitor devices, please supply these data.

Reply: We agree with the reviewer that leakage current and self-discharge time are helpful parameters of supercapacitors. We measured these performances and added them in the revised manuscript. It should be mentioned, according to the suggestions of the reviewer 1, the abbreviation of “AAS” has been replaced with “AHEC” that means aqueous hybrid electrochemical capacitor. As shown in the Fig. R1, the leakage current was tested by charging the AHEC to 1.8 V at a current of 0.04 mA and then keeping the voltage at 1.8 V for 2h. And the leakage current of the AHEC is only 3.5 μA , indicating its good stability. The self-discharge behavior of the AHEC was characterized after charged at 1.8 V for 10 min. The voltage shows a quick decrease in the beginning time. It may be due to the easy ion adsorption/desorption within large porous structure. And then the AHEC can maintain a voltage of 0.16 V after 15000 s. (Please see page 9, line 183–186.)

The Fig. R1 was added in revised Supporting Information as the Fig. S15. (Please see Supplementary Information, page 18.)

Fig. R1. (a) Leakage current curve of the AHEC charged at 0.04 mA to 1.8 V and kept at 1.8 V for 2h. (b) Self-discharge curve of the AHEC after charged at 1.8 V for 10 min.

6. As the author demonstrate a flexible integrated AASs, how is the flexibility and mechanical stability of the devices on working state.

Reply: According to the reviewer’s suggestion, we tested the flexibility and mechanical performance of the integrated AHECs. When the integrated AHECs were bent from 60° to 180°, its CV curves showed a negligible distortion (Fig. R2a). Meanwhile, it also exhibits a good working condition after 1000 bending cycles at a bend angle of 90°, which can be observed by the similar CV curves of the integrated AHECs (Fig. R2b). These confirmed that the integrated AHECs also possess good flexible and mechanical stability on working state. We added the flexibility tests and corresponding description in the revised Supporting Information as the Fig. S22. **(Please see Supplementary Information, page 25.)**

Fig. R2 (a) CV curves of integrated AHECs in at different bending angles. (b) CV curves of integrated AHECs after repeated bending cycles at 90°.

7. There is an AEC used for comparison in this work. Please give detailed technical parameters of it.

Reply: Actually, the basic parameters of the AEC has been described in the main

text “much smaller than those (almost 0.08) generated by the AEC (22 μF / 450 V)” (Please see page 11, line 214.). For clarity, we further added more notes where we mentioned the AEC.

8. For practical application, the author can fabricate solid-state and/or organic AASs?

Reply: According your suggestion, we tested the electrochemical performances of the AHEC unit by using the gel electrolyte (PVA-LiCl) and organic electrolyte (Fig. R3, R4, R5). The PVA-LiCl gel electrolyte prepared according to the previous methods [Chen et al. *Energy Environ. Sci.* **2017**, 10, 538–545]. Specifically, 0.424g LiCl and 1 g PVA were dissolved in 10 g H₂O by continuously stirring at 90 °C. By using the gel electrolyte, the CV curves of AHEC maintain a rectangle shape at different scan rate (from 10 V s⁻¹ to 400 V s⁻¹), also representing the good capacitive behavior (Fig. R3). Electrochemical impedance spectroscopy (EIS) shows AHEC still possesses a relatively low ESR of 0.7 $\Omega \text{ cm}^{-2}$ (Fig. R4a) and a good high-frequency response capability along with a capacitance of 158 $\mu\text{F cm}^{-2}$ and phase angle of -78° at 120 Hz (Fig. R4b,c,d). Meanwhile, the triangular shape of the galvanostatic charge-discharge (GCD) curves exhibits the nearly 100 % coulombic efficiency (Fig. R4e). And after 10,000 cycling test, the capacity retention of the AHEC also maintain 92.7 % (Fig. R4f). These demonstrate that the AHEC could work well with gel electrolyte and have great potential in the practical application. In addition, by using the organic electrolyte, frequency response of AHECs shows a slight decay, and the phase angle at 120 Hz was only 59° (Fig. R5). The degradation of the electrochemical performances may result from the increasing resistance (Fig. R5).

Fig. R1. CV curves of the solid-state AHECs at different scan rates.

Fig. R2. The electrochemical performances of solid-state AHEC. (a) Nyquist plot; inset: the expanded view at high frequencies. (b) Plots of phase angle versus frequency. (c) Plots of areal specific capacitance (C_A) as a function of frequency. (d) Plots of the real or imaginary part of specific capacitance (C' or C'') versus frequency. (e) GCD curves at different current densities. (f) Cycling stability at a current density of 5 mA cm^{-2} for 10,000 cycles.

Fig. R3. The electrochemical performances of organic AHEC unit. (a) Nyquist plot; inset: the expanded view at high frequencies. (b) Plots of phase angle versus frequency. (c) Plots of areal specific capacitance (C_A) as a function of frequency. (d) Plots of the real or imaginary part of specific capacitance (C' or C'') versus frequency.

9. Language and description should be rechecked to make sure correction display. For example, in Page 3, line 3, it is better to call this filter rather than filtering. Page 3, line 15, meet is better than match. Additionally, please rearrange the following sentence, because it looks like uncompleted. Page 6, line 7, “be” should be deleted.

Reply: As you suggested, we have modified the sentences:

(1) “Herein, we reported a wide frequency applicable AC line filter based on an aqueous hybrid electrochemical capacitor (AHEC)” **(Please see page 3, line 50–52.)**

(2) “This voltage window could meet the requirement of electrical appliance from pint-sized portable devices to large-scale ones” **(Please see page 3, line 65–66.)**

(3) “Within higher potential window, the capacitance retention slightly decreased to 84% after 10,000 cycles (Fig. S8), which may due to the decomposition of electrolyte (Fig. S9)²³”. **(Please see page 6–7, line 122–124.)**

In addition, we have double checked the language in the manuscript.

Reviewer #3 (Remarks to the Author):

The authors reported an aqueous asymmetric supercapacitor exhibited excellent electrochemical performances. The cathode and anode synergistically help the supercapacitor realize the efficient conversion of arbitrary AC to DC under an ultra-wide frequency range with hundreds of volts, promising for the renewable power generators. It is a very interesting and important work toward the ultrafast supercapacitors, and greatly broaden the potential application of line-filter capacitors fields. The characterization is also praiseworthy. For these reasons, I believe that this work shall attract wide attention and therefore the manuscript can be accepted for publication in Nature communication after minor revision.

Reply: We would like to thank the reviewer for the positive recommendation. We are also grateful for their helpful and constructive comments to the manuscript. We tried our best to address the concerns in the revised manuscript.

1. In Page 13, line 20, what is the “its” mean?

Reply: “its” means “the capacitors”. And we have changed the terminology of “its” to “the capacitors”. They are marked in red in the revised manuscript. **(Please see page 13, line 250.)**

2. Please provide the parameters and details of the raw materials including GO and PEDOT.

Reply: In the original manuscript, we have characterized the XPS and Raman spectra of the GO and PEDOT. And we added some new characterizations to manifest the physical and chemical structures of these raw materials including SEM images, infrared spectra, and UV-vis spectra.

Fig. R1. (a) SEM images of GO sheets with an average lateral size of 2-4 μm. (b) Attenuated total reflection Fourier transform infrared (FTIR) spectra of GO. (c) UV-vis spectra of GO.

Physical and chemical information of the GO sheets were given by Fig. R1. A relatively small size of 2-4 μm for GO sheets could be observed by the SEM images, which is advantageous to uniform growth of ErGO sheets during electrochemical deposition process [Chi et al. *Adv. Energy Mater.* **2017**, 7, 1700591]. ATR-FTIR spectral studies show the oxygen groups located on the GO sheets including C=O ($1740\text{--}1720\text{ cm}^{-1}$), C–O–C ($\sim 1000\text{ cm}^{-1}$), C–O, (1230 cm^{-1}), and –OH ($3600\text{--}3300\text{ cm}^{-1}$) [Chen et al. *Chem. Sci.* **2016**, 7, 1874–1881], which are also confirmed by XPS analysis. And UV-vis adsorption spectrum of GO exhibits a main adsorption peak at around 230 nm from the $\pi\text{--}\pi^*$ transition of conjugated ketones or dienes. The wide adsorption region between 270 and 600 nm belongs to the conjugated aromatic domains [Eigler et al. *Adv. Mater.* **2013**, 25, 3583–3587]. This figure was added in the revised Supporting Information as the Fig. S28. (Please see **Supplementary Information page 32.**)

Fig. R2. (a) FTIR spectra of PH 1000. (b) UV-vis spectra of PH 1000.

Fig. R2. Shows the basic chemical structures of the PH1000, which is the precursor of PEDOT positive electrode materials. The ATR-FTIR spectra provide more detailed information of the stretching vibrations of benzene sulfonate ($1172, 1128, 1035,$ and 1005 cm^{-1}), indicating the existence of PSS [Yao et al. *Adv. Mater.* **2017**, 29, 1700974]. In addition, in the UV-vis spectra, the higher energy peak at 340 nm is assigned to the $n\text{--}\pi^*$ transition in the PEDOT backbone and the broad band above 600 nm corresponds to bipolaron subgap states of the sample. This figure was added in the revised Supporting Information as the Fig. S30. (Please see **Supplementary Information, page 34.**)

3. The electrical conductivity is important for AC filtering. How is the electrical conductivity measured? Please describe the details in the experimental section.

Reply: The electrical conductivity is measured by the four-point probe KDY-1 sheet resistivity tester (Kunde Technology, Guangzhou, China). The specific method is that the four-point probe is slightly contacted on the graphite foil (or a PEDOT film on glass). Then, turn on the constant current source switch and read out the voltage (V)

and current (I). According to the equation (R1), the sheet resistivity (R_s) could be calculated.

$$R_s = V/I \times F_{(S/D)} \times F_{sp}/ F_t \quad (R1)$$

where $F_{(S/D)}$ is diameter coefficient, F_{sp} is Probe spacing correction coefficient, and F_t is the temperature coefficient.

Thereafter, the electrical conductivity (σ) is calculated by the equation (R2):

$$\sigma = 1/(R_s \times d) \quad (R2)$$

d is the thickness of film, which is measured by SEM images and a P-7 stylus profiler (KLA-Tencor, USA).

We added the description into the characterization section, and they were marked in red in revised manuscript. (Please see page 20, line 404–407.)

4. For the preparation process, whether the ErGO electrode was washed in water after deep further deep reduction? Any influence?

Reply: As we described in the preparation process, the ErGO were fabricated by two steps. The first step is electrochemical deposition process of the ErGO, and the second step is the further reduction of ErGO process. Both the processes were performed in the solution system, containing salt ions (Li^+ and ClO_4^-) that may be absorbed on the surface of ErGO. In order to remove the influences of these salt ions, the ErGO were repeatedly soaked with water after each treatment. The descriptions in the methods seem to be unclear. For clarity, we added the washing step after the deep reduction treatment in the Methods part, “Finally, the ErGO negative electrode was obtained by drowning it in water for several times to remove the residual salt ions within electrode.”. (Please see page 17, line 342–343.)

5. The ultrawide frequency is valuable for practical applications. The frequency range should be checked. For example, the AAS is developed for with an applicable ultrawide frequency range from 1 to 1,000,000 Hz (Fig. 5a). But authors reported “from 10 Hz to 1,000,000 Hz. Furthermore” (Page 3, line 13).

Reply: The wide frequency range is displayed in both single device and the integrated units. Their lowest frequencies are slightly different. For the single device, it can convert the AC to DC with a frequency of 1 Hz. But for the integrated units, the lowest frequency is 10 Hz. It may be due to the capacitance reduction and resistance

increased after series, though RC time is nearly unchanged. And we checked the full text to make sure described the frequency ranges exactly. They are marked in red in the revised manuscript. **(Please see page 1, line 16–18 and page 3, line 60–62.)**

6. The properties and functions of PEDOT and graphene acting as electrode should be briefly discussed in introduction.

Reply: In this manuscript, the PEDOT and ErGO were the positive electrode materials and negative electrode materials, respectively. It is because their good electrochemical properties and conductivities. The two materials have good electrical conductivity (PEDOT film: 1000-2000 S cm⁻¹ Xia et al. *Adv. Mater.* **2012**, 24, 2436–2440 and rGO film: ~1100 S cm⁻¹ Wen et al. *Adv. Mater.* **2017**, 29, 1702831), which are in favor of the quick electron transport within the electrodes. Meanwhile, the two materials possess large porous three-dimension structure which provides the unblocked path and large surface for the ion adsorption/desorption. These promise that these materials can be used as the ideal electrode with low resistance and good capacitance for constructing the fast response electrochemical capacitors. And we have added the brief discussion about the properties of the electrode materials in the introduction part, “In this EC, both the stable electrode materials have high conductivities and large porous structures, being in favor of quick electron transport and good electric double layer capacitive behaviors”. **(Please see page 3, line 54–57.)**

7. It is better to improve the use of English language throughout the manuscript. For example, in Page 12 line 3, “a sinusoidal AC signal was firstly converted into one constant polarity signal by flowing through a bridge rectifier...”; in Page 13, line 3 “the rectifying signals was further smoothed...”

Reply: As you suggested, we have modified the sentences

(1) “For the typical filtering process, a sinusoidal AC signal was firstly converted into one constant polarity signal after flowing through a bridge rectifier which is composed of four diodes (Fig. 5c, e; Fig. S18).” **(Please see page 12, line 227–230.)**

(2) “Subsequently, the rectifying signals were further smoothed by the filtering capacitor to produce...” **(Please see page 14, line 233–234.)**

We have also double checked and polished the English spelling and grammar of the manuscript.

Reviewers' comments:

Reviewer #1 (Remarks to the Author):

The revised manuscript is improved over the original manuscript but it remains unacceptable for publication. The major problem is that the technology described is not able to efficiently filter ac to dc over an ultra-wide frequency range, which is contrary to manuscript emphasis and statements that are made.

This work is interesting and it could become acceptable for publication with a major change in emphasis. This new emphasis could be, for instance, the cell fabrication approach including the approach to series-connect cells with performance results. Other emphasis selections also appear possible.

To assist the authors when they prepare a new manuscript, Reviewer #1 responded to several of the author rebuttal comments, which are listed below:

1) The word "supercapacitor" was changed in many places in the revised manuscript as requested but this change was not made in lines 39, 43, 44, 45, 60, 144, 167, 174, 180, and in other places.

2) ok

3) The operating characteristics of any two-terminal electrical component are determined by its complex impedance. A component behaves like a capacitor in the frequency region over which its phase angle is negative and that same component behaves like an inductor in the frequency region over which its phase angle is positive. The self-resonant frequency is defined as the frequency at which the value of the phase angle is exactly zero. Figure 3 in the revised manuscript is a phase angle plot of one AHEC unit and Figure 6 is a phase angle plot of multiple AHEC units connected in series. Both plots show a self-resonant frequency of approximately 20 kHz. Thus AHEC units (one or multiple in series) will operate as capacitors below approximately 20 kHz and at higher frequency, operate as inductors. The units described do not have ultra-wide frequency characteristics but rather behave like some aluminum electrolytic capacitors. The given explanation about filtering over an ultra-wide frequency range that invokes an electric field argument is irrelevant. This conclusion is not speculative—its basis can be found in any elementary circuit engineering textbook.

Note that any capacitor can be charged using a 1 MHz rectified waveform because there is an average positive voltage. Such charging does not in any way demonstrate filtering.

Reviewer #2 (Remarks to the Author):

The author has addressed all my concerns, hence, this paper can be published in its current form.

Reviewer #3 (Remarks to the Author):

The manuscript is significantly improved. I therefore recommend its publication.

Point-by-Point Responses to the Reviewer's Comments

Reviewer #1 (Remarks to the Author):

The revised manuscript is improved over the original manuscript but it remains unacceptable for publication. The major problem is that the technology described is not able to efficiently filter ac to dc over an ultra-wide frequency range, which is contrary to manuscript emphasis and statements that are made.

This work is interesting and it could become acceptable for publication with a major change in emphasis. This new emphasis could be, for instance, the cell fabrication approach including the approach to series-connect cells with performance results. Other emphasis selections also appear possible.

Reply: We are sincerely grateful for the kind suggestions. Accordingly, we have adjusted our emphasis in the revised manuscript with focus more on the high voltage cells configuration and performances, as well as cell fabrication approach. The reported data and results represent a wide interest and importance in the relevant fields. In addition, the frequency range of 1 to 10,000 Hz for the AC filtering is still the widest reported in the published literatures, yet we don't overemphasize that. **(Please see main text line 1, 15–17, 41–43, 48–49, 59–61, 208–213, 257–259 274–275, 289–291, 299–304, Figure 5a,b, and Figure 6f. Supporting Information line 1, 213–220, Figure S27 Table S2)**

To assist the authors when they prepare a new manuscript, Reviewer #1 responded to several of the author rebuttal comments, which are listed below:

1) The word “supercapacitor” was changed in many places in the revised manuscript as requested but this change was not made in lines 39, 43, 44, 45, 60, 144, 167, 174, 180, and in other places.

Reply: As you suggested, we have changed all the word “supercapacitor”. **(Please see main text line 37, 41, 43, 58, 141, 163, 171, 177, 213, 268, and 285. Supporting Information line 59, 60, 337, 339, and 343)**

2) ok

Reply: Thanks for your comment.

3) The operating characteristics of any two-terminal electrical component are determined by its complex impedance. A component behaves like a capacitor in the frequency region over which its phase angle is negative and that same component behaves like an inductor in the frequency region over which its phase angle is positive.

The self-resonant frequency is defined as the frequency at which the value of the phase angle is exactly zero. Figure 3 in the revised manuscript is a phase angle plot of one AHEC unit and Figure 6 is a phase angle plot of multiple AHEC units connected in series. Both plots show a self-resonant frequency of approximately 20 kHz. Thus AHEC units (one or multiple in series) will operate as capacitors below approximately 20 kHz and at higher frequency, operate as inductors. The units described do not have ultra-wide frequency characteristics but rather behave like some aluminum electrolytic capacitors. The given explanation about filtering over an ultra-wide frequency range that invokes an electric field argument is irrelevant. This conclusion is not speculative—its basis can be found in any elementary circuit engineering textbook. Note that any capacitor can be charged using a 1 MHz rectified waveform because there is an average positive voltage. Such charging does not in any way demonstrate filtering.

Reply: Thanks a lot for the very helpful and instructive comments. According to your suggestions, we have edited the description about excessively high frequency range and modified the corresponding figures. Meanwhile, the emphasis in the manuscript has been adjusted. And we also added the discussion about self-resonant frequency in the revised manuscript as you mentioned in this comment. The statement is “Both the AEC and AHEC exhibits a typical properties of RCL circuit including the capacitive impedance ($1/2\pi fC$), inductive impedance ($2\pi fL$), and ESR. The ESR could be read at the frequency that $1/2\pi fC = 2\pi fL$. It is also the lowest point in the curves. And at this point, the corresponding frequency is known as self-resonant frequency (SRF). For the two-terminal electrical component, the SRF is the separation determining that the component behaves like a capacitor or an inductor. In our AHEC and the AEC, the SRF is approximately 20 kHz. Thus, below the 20 kHz, they can be used as the filtering capacitor.” (Please see main text line 1, 15–17, 41–43, 48–49, 59–61, 208–213, 257–259 274–275, 289–291, 299–304, Figure 5a,b, and Figure 6f. Supporting Information line 1, 213–220, Figure S27 Table S2)

Reviewer #2 (Remarks to the Author):

The author has addressed all my concerns, hence, this paper can be published in its current form.

Reply: Thanks for the supportive comment.

Reviewer #3 (Remarks to the Author):

The manuscript is significantly improved. I therefore recommend its publication.

Reply: Thanks for the supportive comment.

List of changes in main text

1. Page 1, line 1 the word “ultra” was deleted and the title was modified to “Wide Frequency and Arbitrary Waveform Applicable AC Line Filtering up to Hundreds of Volts Based on Aqueous Hybrid Electrochemical capacitors”
2. Page 1, line 17, a new sentence described the frequency range, “Herein, an aqueous hybrid electrochemical capacitor (AHEC) is developed for alternate current line filtering with an applicable wide frequency range from 1 to 10,000 Hz.”.
3. Page 2, line 37, 41, 43 the abbreviation and words “Supercapacitors (SCs)” were revised to “Electrochemical capacitors (ECs)”.
4. Page 2, line 41–43, the emphasis on frequency was replaced by emphasis on voltage, and a new sentence is “Nevertheless, an applicable ECs for practical filtering is still lacking, mostly because of their relatively low working voltage”.
5. Page 3, line 48–49, a new emphasis on the series-connected configuration with hundreds of volts was added, the sentence is “Herein, we reported a hundreds of volts workable AC line filter based on series-connected configuration of aqueous hybrid electrochemical capacitors (AHECs)”.
6. Page 3, line 58, the abbreviation “SCs” was revised to ECs”.
7. Page 3, line 59–61, the frequency range has been adjusted. The revised sentence is “Consequently, the large specific energy density and low total impedance of this AHEC enables the efficient smoothing of AC into DC within a wide frequency range from 1 Hz to 10,000 Hz”.
8. Page 7, line 141, the abbreviation “SC” was revised to “EC”.
9. Page 9, line 163, 171, and 177, the abbreviation “SC” was revised to “EC”.
10. Page 11, line 208–213 the frequency range has been adjusted. The revised sentences are “This AHEC unit works well with a frequency range from 1 Hz to 10,000 Hz (Fig. 5a; Fig. S17). In such a broad frequency range, all the output signals maintain a smooth line shape with a negligible variance of less than 0.01, much smaller than those (almost 0.08) generated by the AEC (22 μ F/ 450 V) (Fig. 5b). Meanwhile, the applicable frequency range from 1 Hz to 10,000 Hz is the widest scope achieved to date for ECs, surpassing that of all the reported ultra-fast ECs (Table S2)”.
11. Page 11, line 213, the abbreviation “SC” was revised to “EC”.
12. Page 12, Figure 5a,b, the frequency range has been revised below the self-resonant frequency.
13. Page 14, line 257–259, a new sentence described the self-resonant frequency, “Therefore, the AHEC unit demonstrates a great promise for AC/DC conversion within frequency range below the self-resonant frequency (Fig. S19)”.

14. Page 14, line 259, the previous description about the construction of the electric field has been deleted, “~~In addition, it should be mentioned, when the frequency is higher than the self resonant frequency (SFR) at which the capacitive impedance ($\frac{1}{2\pi fC}$) equals the inductive impedance ($2\pi fL$), the capacitor (including the AHEC and AEC) can also achieve the ripple filtering. That may be because the electrical field of the capacitors can be constructed even though the arisen inductive behavior would limit them (Fig. S20).~~”
15. Page 14, line 268, the abbreviation “SC” was revised to “EC”.
16. Page 14, Figure 6f, the frequency range has been adjusted, and the 10–10,000 Hz is still the widest frequency range among the reported AC-line filtering ECs.
17. Page 15, line 274–275, the new sentence that described the connected manner of the AHEC units has been added “More specifically, the AHEC unit was connected by the exposed graphite foils which act as wires and are fixed by polyimide tapes”.
18. Page 15, line 285, the abbreviation “SC” was revised to “EC”.
19. Page 15, line 289–291, the frequency range has been adjusted. The revised sentence is “Moreover, the planar AHEC integration can work at much high voltage (e.g., 220 V_{peak-peak} and 400 V_{peak-peak}) within a wide frequency range (10 Hz to 10,000 Hz, Fig. S27)”.
20. Page 16, line 299–304, the frequency range has been adjusted. The revised sentences are “A unique AHEC was developed for AC-line filtering with high specific capacitance and enlarged voltage window, which work well with a wide frequency range from 1 Hz to 10,000 Hz and an accumulative voltage of hundreds of volts. Meanwhile, the AHEC displays a record high areal specific energy density of 438 $\mu\text{F V}^2 \text{ cm}^{-2}$ at 120 Hz, and can filter arbitrary AC waveforms to smooth DC within wide frequency range and high voltage condition.”

List of changes in supporting information

1. Page 1, line 1 the word “ultra” was deleted and the title was modified to “Wide Frequency and Arbitrary Waveform Applicable AC Line Filtering up to Hundreds of Volts Based on Aqueous Hybrid Electrochemical capacitors”
2. Page 3, line 59, 60, the abbreviation “SC” was revised to “EC”.
3. Page 22, line 213–220, the discussion about the self-resonant frequency was added, “Both the AEC and AHEC exhibits a typical properties of RCL circuit including the capacitive impedance ($1/2\pi fC$), inductive impedance ($2\pi fL$), and ESR. The ESR could be read at the frequency that $1/2\pi fC = 2\pi fL$. It is also the

lowest point in the curves. And at this point, the corresponding frequency is known as self-resonant frequency (SRF). For the two-terminal electrical component, the SRF is the separation determining that the component behaves like a capacitor or an inductor. In our AHEC and the AEC, the SRF is approximately 20 kHz. Thus, below the 20 kHz, they can be used as the filtering capacitor.”

4. Page 23, previous Figure S20 which described the construction of the electric field at high frequency has been removed.
5. Page 30, Figure S27 has been modified.
6. Page 34, line 319, the abbreviation “SC” was revised to “EC”.
7. Page 35, line 337, 339, and 343, the abbreviation “SC” was revised to “EC”.
8. Page 35, Table S2, the frequency range has been adjusted.

REVIEWERS' COMMENTS:

Reviewer #1 (Remarks to the Author):

This revision is greatly improved over the earlier manuscript. It should be ready for publication after several small changes are made. These include:

1. Data from references S8 and S9 should be added to Supplementary Table 2. Both of these references show data with self-resonant frequencies higher than values reported in the manuscript capacitors. Thus, the sentence in lines 214-216 should be deleted since the claim made is not true, and the sentence on lines 300-303 should also be modified to make it true.
2. The sentence on lines 185-186 is the last sentence in this paragraph. This sentence does not follow from the development in the paragraph (not supported) and it should be deleted.
3. On line 345, the word "drowning" should be changed to "soaking". This will improve the English. Why? Because the word drowning is typically associated with living organisms while the word soaking is associated with anything, alive or inanimate.

Point-by-Point Responses to the Reviewers' Comments

Reviewer #1 (Remarks to the Author):

This revision is greatly improved over the earlier manuscript. It should be ready for publication after several small changes are made. These include:

Reply: Great thanks for the reviewer's positive comments.

1. Data from references S8 and S9 should be added to Supplementary Table 2. Both of these references show data with self-resonant frequencies higher than values reported in the manuscript capacitors. Thus, the sentence in lines 214-216 should be deleted since the claim made is not true, and the sentence on lines 300-303 should also be modified to make it true.

Reply: According to your suggestions, the data from the references S8 and S9 have been added to the Fig. 6f and Supplementary Table S2. Meanwhile, the sentences which describe the wide frequency range have been deleted and modified. The revised sentence is "This report of filtering capacitors simultaneously achieves the relatively wide frequency range and high adapted voltage on demand without any sacrificing its filtering performance." (Please see main text line 306–308, Fig. 6f and Supplementary Table S2)

2. The sentence on lines 185-186 is the last sentence in this paragraph. This sentence does not follow from the development in the paragraph (not supported) and it should be deleted.

Reply: Thank you for your kind suggestion! Accordingly, we have deleted this irrelevant sentences to make the expression smoother.

3. On line 345, the word "drowning" should be changed to "soaking". This will improve the English. Why? Because the word drowning is typically associated with living organisms while the word soaking is associated with anything, alive or inanimate.

Reply: Thanks a lot for your kind comments! The word "drowning" has been changed to "soaking" in the revised manuscript. (Please see main text line 350)